# Max Entropy Moment Kalman Filter for Polynomial Systems with Arbitrary Noise

**Sangli Teng**[*]
sanglit@umich.edu

**Harry Zhang**[†‡]
harryz@mit.edu

**David Jin**[†‡]
jindavid@mit.edu

**Ashkan Jasour** [§]
jasour@jpl.caltech.edu

**Ram Vasudevan**[*]
ramv@umich.edu

**Maani Ghaffari**[*]
maanigj@umich.edu

**Luca Carlone**[†]
lcarlone@mit.edu

## Abstract

Designing optimal Bayes filters for nonlinear non-Gaussian systems is a challenging task. The main difficulties are: 1) representing complex beliefs, 2) handling non-Gaussian noise, and 3) marginalizing past states. To address these challenges, we focus on polynomial systems and propose the *Max Entropy Moment Kalman Filter* (MEM-KF). To address 1), we represent arbitrary beliefs by a Moment-Constrained Max-Entropy Distribution (MED). The MED can asymptotically approximate almost any distribution given an increasing number of moment constraints. To address 2), we model the noise in the process and observation model as MED. To address 3), we propagate the moments through the process model and recover the distribution as MED, thus avoiding symbolic integration, which is generally intractable. All the steps in MEM-KF, including the extraction of a point estimate, can be solved via convex optimization. We showcase the MEM-KF in challenging robotics tasks, such as localization with unknown data association.

## 1 Introduction

The Kalman Filter (KF) is an optimal estimator for linear Gaussian systems in the sense that it provably computes a minimum-variance estimate of the system's state. From the perspective of the Bayes Filter, the KF recursively computes a Gaussian distribution that captures the mean and covariance of the state. However, the optimality of the KF breaks down when dealing with nonlinear systems or non-Gaussian noise. A plethora of extensions of the KF has been developed to deal with nonlinearity and non-Gaussianity. For example, linearization-based methods, such as the Extended Kalman Filter (EKF) [1], linearize the process and measurement models and propagate the covariance using a standard KF. Other techniques, such as the Unscented Kalman Filter (UKF) [2], rely on deterministic sampling to better capture the nonlinearities of the system but then still fit a Gaussian distribution to the belief. Unfortunately, these extensions do not enjoy the desirable theoretical properties of the KF: they are not guaranteed to produce an optimal estimate and —in the nonlinear or non-Gaussian case— their mean and covariance might be far from describing the actual belief distribution. Consider for example a system with non-Gaussian noise sampled from a discrete

---

[*]University of Michigan, Ann Arbor, MI 48109, USA

[†]Massachusetts Institute of Technology, Cambridge, MA, 02139, USA

[‡]Equal contributions

[§]Team 347T-Robotic Aerial Mobility, Jet Propulsion Lab, Pasadena, CA, 91109, USA

39th Conference on Neural Information Processing Systems (NeurIPS 2025).

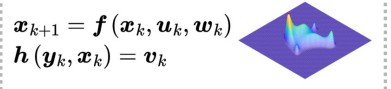
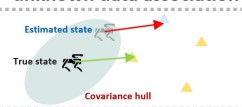

$$\begin{aligned} \boldsymbol{x}_{k+1} &= \boldsymbol{A}\boldsymbol{x}_k + \boldsymbol{w}_k \\ \boldsymbol{y}_k &= \boldsymbol{C}\boldsymbol{x}_k + \boldsymbol{v}_k \end{aligned}$$

$$\begin{aligned} \boldsymbol{x}_{k+1} &= \boldsymbol{f}\left(\boldsymbol{x}_k, \boldsymbol{u}_k, \boldsymbol{w}_k\right) \\ \boldsymbol{h}\left(\boldsymbol{y}_k, \boldsymbol{x}_k\right) &= \boldsymbol{v}_k \end{aligned}$$

Estimated state
True state
Covariance hull

Figure 1: We generalize the classical Kalman Filter (KF) by proposing the Max Entropy Moment Kalman Filter (MEM-KF), which operates on a polynomial system with arbitrary noise. In the Bayes Filter framework, the KF predicts and updates a Gaussian distribution that can be characterized by its mean and covariance. The MEM-KF predicts and updates a max entropy distribution that can be uniquely characterized by its higher-order moments. The proposed algorithm is evaluated in a robot localization problem with unknown data association. In this case, the robot does not know which landmark it observes, which can be modeled by a multi-modal distribution.

distribution over $\{-1, 1\}$; clearly, approximating this noise as a zero-mean Gaussian fails to capture the bimodal nature of the distribution. For this reason, existing nonlinear filters, such as the EKF and the UKF, can hardly capture the noise distribution precisely with Gaussian assumptions.

Thus, a natural question is, *how can we provide provably optimal state estimation for nonlinear systems corrupted by arbitrary noise?* When the noise is non-Gaussian, and the system is nonlinear, the estimation problem becomes less structured for analysis, and traditional methods only focus on the first two moments (i.e., mean and covariance) to describe the noise and the belief, which are too crude to capture the true underlying distributions. We argue that by accounting for higher-order moments, we can recover the state distribution more precisely. In this work, we focus on *polynomial systems*, a broad class of potentially nonlinear systems described by polynomial equations (Fig. 1). These systems can model dynamics on matrix Lie groups [3] and a broad set of measurement models [4]. To handle arbitrary noise, we let the user specify an arbitrary number of moments to better describe the process and measurement noise.

**Contribution.** We propose the *Max Entropy Moment Kalman Filter* (MEM-KF) with three key contributions. Our first contribution is to formulate the state estimation problem as a Kalman-type filter using the Moment-Constrained Max-Entropy Distribution (MED) as the belief representation. The MED can asymptotically approximate any moment determinate distribution. Given the moments, the coefficients of the MED can be recovered by convex optimization.

Our second contribution is to formulate the prediction and update step of the MEM-KF and solve them via convex optimization. In the prediction step, we propagate the moments through the process model and recover the belief as MED. In the update step, we formulate the sensor model as MED and show how to obtain the posterior distribution.

Our third contribution is to extract the optimal point estimate via convex optimization. Due to the polynomial nature of the MED, we apply semidefinite moment relaxation techniques to extract the optimal point estimate given the belief representation. The global optimality of the solution (resp. a non-trivial suboptimality bound) can be certified by using a rank condition (resp. the relaxation gap) of the moment relaxation. We showcase the resulting MEM-KF in challenging robotics tasks, such as localization with unknown data association.

Compared to the traditional marginalization step in the Bayes Filter that requires symbolic integrations and is generally intractable, the proposed method only requires numerical integrations to evaluate the gradients for the convex optimization. Although the numerical integration is still relatively expensive, MEM-KF provides a general solution for filtering in polynomial systems with arbitrary noise.

## 2 Related Work

**Bayes Filter.** The Bayes filter recursively estimates the posterior probability of a system's state by predicting the prior from the process model and updating it with new measurements using Bayes' rule [5]. For a nonlinear process model, even a simple distribution can be skewed and transformed to a multi-model distribution that is hard to approximate. As a solution, the prior can be approximated by parametric functions, such as a Gaussian mixture [6, 7]. Despite that, the belief can be asymptotically approximated with larger Gaussian sums, the prediction still assumes linear process models [6] or relies on linearization [7]. Other approximation of belief includes point mass [8, 9] and piece-wise constant [10, 11] that are nonsmooth. Contrary to the deterministic method that approximates the

prior by parametric functions, sampling-based methods such as particle filters [12] have been applied to approximate the predicted distribution. Though the sampling-based methods are more versatile for nonlinear non-Gaussian systems, these methods generally lack formal guarantees. We recommend the readers to [5] for a more comprehensive review.

**Kalman Filtering.** When the process and measurement models are linear, and the noises are Gaussian, the Kalman Filter [13] becomes provably optimal and returns the minimum-variance estimate among all estimators. To deal with nonlinear systems, the Extended Kalman Filter (EKF) linearizes the process and measurement models around the estimated states [14] and propagates the covariance through the linearized system. However, the performance of EKF deteriorates as the linearization points deviate from the true states, as the process model becomes more nonlinear and the noise increases [1, 15]. To tackle this problem, symmetry-preserving linearization has been applied in homogeneous space [16–18] that can avoid the inconsistency caused by imperfect state estimations. As an alternative to linearizations, the *Unscented Kalman Filter (UKF)* [2] applies a deterministic sampling approach to handle nonlinearities. However, for non-Gaussian noise, UKF lacks a systematic way to handle arbitrary noise. We note that both EKF and UKF adopt the Kalman gain and only use the mean and covariance of the noise distribution. Shimizu et al. [19] and Jasour et al. [20] are closer to our work, where moment propagation is derived for filtering problems by assuming knowledge of the characteristic function of the Gaussian noise. In our methods, we only utilize moment information instead of the entire distribution, e.g., the characteristic function, which is inaccessible in typical applications. Though [21] utilized higher-order moments to derive a minimal variance filter, the prediction applies approximations that is not exact.

**Semidefinite Relaxation** Semidefinite relaxations have been critical to designing certifiable algorithms for robot state estimation. These relaxations transform polynomial optimization problems into semidefinite programs (SDPs) that are convex, using Lasserre's hierarchy of moment relaxations [22]. Subsequently, SDP relaxation-based certifiable algorithms have been extended to various geometric perception problems, including pose graph optimization [23, 24], rotation averaging [25, 26], triangulation [27, 28], 3D registration [29–32], absolute pose estimation [33], and category-level object perception [34, 35]. Compared to belief, e.g., [6–11] that are hard to extract or certify the globally optimal point estimation, the optimal point estimation of MED can be solved by SDPs.

## 3 Preliminaries

We review key concepts in polynomial systems. Let $\mathbb{R}[\boldsymbol{x}]$ be the ring of polynomials with real coefficients, where $\boldsymbol{x} := [x_1, x_2, \ldots, x_n]^\mathsf{T}$. Given an integer $r$, we define the set $\mathbb{N}_r^n := \{\alpha \in \mathbb{N}^n \mid \sum_i \alpha_i \le r\}$ (i.e., $\alpha$ is a vector of integers that sums up to no more than $r$). A monomial of degree up to $r$ is denoted as $\boldsymbol{x}^\alpha := x_1^{\alpha_1} x_2^{\alpha_2} \ldots x_n^{\alpha_n}, \alpha \in \mathbb{N}_r^n$. For a polynomial $p(\boldsymbol{x}) := \sum_\gamma c_\gamma \boldsymbol{x}^\gamma$, its degree $\deg p$ is defined as the largest $\|\gamma\|_1$ with $c_\gamma \ne 0$.

Let $s(r, n) := |\mathbb{N}_r^n| = \binom{n+r}{n}$. We then define the basis function $\boldsymbol{\phi}_r(\boldsymbol{x}) : \mathbb{R}^n \to \mathbb{R}^{s(r,n)}$ using all the entries of the canonical basis $\boldsymbol{\phi}_r(\boldsymbol{x}) = [1, x_1, x_2, \ldots, x_n, x_1^2, x_1 x_2, \ldots, x_n^2, x_1^r, \ldots, x_n^r]^\mathsf{T}$.

Given a probability distribution $p(\boldsymbol{x})$ over support $\mathcal{K}$, we can compute the moment matrix of probability $p(\boldsymbol{x})$ as $\boldsymbol{M}_r(\bar{\boldsymbol{x}}_\alpha) = \int_\mathcal{K} \boldsymbol{M}_r(\boldsymbol{x}) p(\boldsymbol{x}) d\boldsymbol{x}, \alpha \in \mathbb{N}_{2r}^n$, where we have $\bar{\boldsymbol{x}}_\alpha := \int_\mathcal{K} \boldsymbol{x}^\alpha p(\boldsymbol{x}) d\boldsymbol{x}$ to denote the moment of $\boldsymbol{x}^\alpha$. The inverse of this process is the moment problem that tries to identify the distribution given the moment sequence. A special case is the moment-determinate distribution that describes a wide range of well-behaved probabilities that are neither long-tailed nor have higher-order moments growing too fast:

**Definition 1** (Moment-determinate distribution). *A distribution $p(\boldsymbol{x})$ is said to be moment-determinate if it can be uniquely determined by all the moments with order from $0$ to $\infty$.*[5]

We introduce the following equality-constrained Polynomial Optimization Problem (POP) that will be applied to extract the optimal point estimation given an MED:

---

[5]With finite order $r$, there can be an infinite number of distributions with the specified moments.

**Definition 2** (Equality-constrained POP). *An equality-constrained polynomial optimization problem (POP) is an optimization problem with polynomial cost and constraints $c(\boldsymbol{x}), g(\boldsymbol{x}) \in \mathbb{R}[\boldsymbol{x}]$:*

$$c^* := \inf_{\boldsymbol{x}} c(\boldsymbol{x}) \quad \text{s.t.} \quad g_j(\boldsymbol{x}) = 0, \quad \forall j \in \{1, \ldots, m\}, \tag{POP}$$

The moment relaxation that solve (POP) via SDPs is presented in Appendix A. Some introductory examples of the moment matrix and polynomial systems are provided in Appendix B and C. In the perception problems, such relaxed SDPs usually retrieve globally optimal solutions to the original (POP), meaning that the relaxation is often exact [28, 36, 37].

## 4  Problem Formulation

We consider the following polynomial dynamical system:

$$\begin{cases} \boldsymbol{x}_{k+1} = \boldsymbol{f}(\boldsymbol{x}_k, \boldsymbol{u}_k, \boldsymbol{w}_k) \\ \boldsymbol{h}(\boldsymbol{y}_k, \boldsymbol{x}_k) = \boldsymbol{v}_k \end{cases}, \qquad \boldsymbol{x}_k \in \mathcal{K}, \ \forall\, k, \tag{1}$$

where $\boldsymbol{x}_k$ is the deterministic state to estimate, $\boldsymbol{y}_k$ are the measurements, and $\boldsymbol{u}_k$ is the control input, all defined at discrete time $k$. We assume that the state $\boldsymbol{x}_k$ is restricted to the domain $\mathcal{K}$, e.g., the set of 2D poses. Both the process model $\boldsymbol{f}$ and the observation model $\boldsymbol{h}$ are vector-valued real polynomials. We take the standard assumption that the process noise $\boldsymbol{w}_k$ and measurement noise $\boldsymbol{v}_k$ are identically and independently distributed across time steps. For polynomial systems, we make the following assumption:

**Assumption 1** ($\mathcal{K}$). *The domain $\mathcal{K}$ is either the Euclidean space $\mathcal{K} = \mathbb{R}^n$ (i.e., the state is unconstrained), or can be described by polynomial equality constraints $\mathcal{K} = \{\boldsymbol{x} | \boldsymbol{g}(\boldsymbol{x}) = 0 \in \mathbb{R}^m\}$.*

Assumption 1 is relatively mild and captures a broad set of robotics problems where the variables belong to semi-algebraic sets (e.g., rotations, poses); see, for instance [38, 39]. Given the system in eq. (1), we formally define the filtering problem that recursively estimates the belief $p(\boldsymbol{x})$:

**Problem 1** (Recursive State Estimation). *Assume the state at time $k + 1$ is solely dependent on the previous state $\boldsymbol{x}_k$, action $\boldsymbol{u}_k$, and measurement $\boldsymbol{y}_k$, we have the **process** and **observation** model as:*

$$p(\boldsymbol{x}_{k+1} | \boldsymbol{x}_k, \boldsymbol{u}_k, \boldsymbol{x}_{k-1}, \cdots, \boldsymbol{x}_0) = p(\boldsymbol{x}_{k+1} | \boldsymbol{x}_k, \boldsymbol{u}_k), \qquad p(\boldsymbol{y}_k | \boldsymbol{x}_k, \boldsymbol{y}_k, \cdots, \boldsymbol{x}_0) = p(\boldsymbol{y}_k | \boldsymbol{x}_k). \tag{2}$$

*Our goal is to recursively estimate the belief $p(\boldsymbol{x}_k) := p(\boldsymbol{x}_k | \boldsymbol{u}_k, \cdots, \boldsymbol{u}_0, \boldsymbol{y}_k, \cdots, \boldsymbol{y}_0)$ for the system in* (1) *dependent on all the information from process and observation models until time $k$.*

In the Bayes Filter, we apply the **prediction** and **update** steps to fuse the dynamics and measurements:

$$p^-(\boldsymbol{x}_{k+1}) \propto \int_{\mathcal{K}} p(\boldsymbol{x}_{k+1} | \boldsymbol{x}_k, \boldsymbol{u}_k) p(\boldsymbol{x}_k) d\boldsymbol{x}, \qquad \text{(Prediction)}$$

$$p(\boldsymbol{x}_k) \propto p(\boldsymbol{y}_k | \boldsymbol{x}_k) p^-(\boldsymbol{x}_k). \qquad \text{(Update)}$$

As the system (1) is nonlinear and corrupted by arbitrary noise, solving Problem 1 via Bayes filter is challenging in representing the belief $p(\boldsymbol{x})$ and the noise. When non-trivial noise belief is involved, marginalizing the distribution in the prediction steps is also generally intractable. In the following sections, we leverage the polynomial structure of the system to address these challenges.

## 5  Moment-Constrained Max-Entropy Distribution

In this section, we introduce the MED and show how to recover the distribution given the moment constraints using convex optimization.

### 5.1  Asymptotic Property of MED

Though a moment-determinate distribution can be uniquely determined by all its moments, for computational purposes, we expect to describe distributions with a finite number of moments. Given a finite number of moments, the underlying distribution is usually not unique. Therefore, to guarantee uniqueness, we define the notion of *Moment-Constrained Max-Entropy Distribution*,

which is the distribution with maximum entropy among all the distributions that match the given moments. We denote the set of distributions that match a set of given moments of order up to $r$ as $\mathcal{P}_r := \{p(\boldsymbol{x}) \mid \int_{\mathcal{K}} p(\boldsymbol{x})d\boldsymbol{x} = \bar{\boldsymbol{x}}_\alpha, \forall \alpha \in \mathbb{N}_r^n\}$, and its limit as $\mathcal{P}_\infty := \lim_{r \to \infty} \mathcal{P}_r$.

Thus, the MED can be obtained by maximizing the entropy functional over $\mathcal{P}_r$:

$$p_r^*(\boldsymbol{x}) := \arg\max_{p(\boldsymbol{x})} \quad -\int_{\mathcal{K}} p(\boldsymbol{x}) \ln p(\boldsymbol{x})d\boldsymbol{x} \quad \text{s.t.} \quad p(\boldsymbol{x}) \in \mathcal{P}_r. \qquad (P_r)$$

We denote the solution to its limiting problem $(P_\infty)$ with feasible set $\mathcal{P}_\infty$ as $p_\infty^*(\boldsymbol{x})$. For moment-determinate distribution in Definition 1, $p_\infty^*(\boldsymbol{x})$ can be uniquely determined by $\mathcal{P}_\infty$. The asymptotic property of $p_r^*(\boldsymbol{x})$ can be summarized by:

**Theorem 1** (Asymptotic approximation of Moment-Determinate Distribution [40]). *Suppose that $\mathcal{P}_\infty$ is defined by the moments of a moment determinate distribution, the solution $p_\infty^*(\boldsymbol{x})$ is unique and $p_r^*(\boldsymbol{x})$ converges to $p_\infty^*(\boldsymbol{x})$ in norm,[6] i.e., $\lim_{r \to \infty} \|p_r^*(\boldsymbol{x}) - p_\infty^*(\boldsymbol{x})\|_1 = 0$.*

*Proof.* The complete proof is shown in [40]. $\qquad\square$

**Remark 1.** *We consider MED as the belief representation for our filtering approach. The resulting approach avoids the integral in the* (Prediction) *step —which is generally intractable— by directly operating over the moments, and then recovers the corresponding distribution via* $(P_r)$*. The approach guarantees convergence to the true belief in the limit, i.e. $r \to \infty$, thanks to Theorem 1.*

**Remark 2.** *For $p(\boldsymbol{x})$ with compact support, $p(\boldsymbol{x})$ is guaranteed to be moment determinate, i.e., $p_\infty^*(x)$ be unique; see Proposition 12.17 of [41]. For an engineering problem where states are not possible to be infinite, additional assumptions on the compactness of support are reasonable.*

## 5.2 Recover MED via Convex Optimization

Now we briefly review the structure of the solution of $(P_r)$ and how we can retrieve it via convex optimization. The full derivation by [42] in a normed linear space is summarized in Appendix D-E.

By the first-order optimality condition, we can show that the solution to $(P_r)$ takes the following form —with the coefficients $\lambda_\alpha$ to be determined:

$$p(\boldsymbol{x}) = \exp\left(-\sum_\alpha \lambda_\alpha \boldsymbol{x}^\alpha\right). \qquad (3)$$

Via the Legendre-Fenchel dual of $(P_r)$, the optimal coefficients can be obtained by minimizing the thermal dynamics potential $\Delta(\boldsymbol{\lambda})$ of $p(\boldsymbol{x})$ [42]:

$$\min_{\boldsymbol{\lambda}} \quad \Delta(\boldsymbol{\lambda}) := \int_{\mathcal{K}} \exp\left(-\sum_\alpha \lambda_\alpha \boldsymbol{x}^\alpha\right)d\boldsymbol{x} + \sum_\alpha \lambda_\alpha \bar{\boldsymbol{x}}_\alpha. \qquad (4)$$

We observe the gradients,

$$\frac{\partial \Delta(\boldsymbol{\lambda})}{\partial \lambda_\beta} = -\int_{\mathcal{K}} \exp\left(-\sum_\alpha \lambda_\alpha \boldsymbol{x}^\alpha\right)\boldsymbol{x}^\beta d\boldsymbol{x} + \bar{\boldsymbol{x}}_\beta \in \mathbb{R}^{|\mathbb{N}_r^n|} \qquad (5)$$

equal zero when $p(\boldsymbol{x})$ satisfies all the moment constraints. Thus, we minimize $\Delta(\boldsymbol{\lambda})$ to search $p(\boldsymbol{x})$ that meets the first-order optimality constraints. We can further verify the convexity of $\Delta(\boldsymbol{\lambda})$ by verifying that the following Hessian is positive definite:

$$\boldsymbol{H}(\boldsymbol{\lambda})_{\beta,\gamma} := \frac{\partial \Delta(\boldsymbol{\lambda})}{\partial \lambda_\beta \lambda_\gamma} = \int_{\mathcal{K}} \exp\left(-\sum_\alpha \lambda_\alpha \boldsymbol{x}^\alpha\right)\boldsymbol{x}^\beta \boldsymbol{x}^\gamma d\boldsymbol{x} \in \mathbb{R}^{|\mathbb{N}_r^n| \times |\mathbb{N}_r^n|}.$$

We find that $\boldsymbol{H}(\boldsymbol{\lambda})$ is exactly the moment matrix generated by $p(\boldsymbol{x})$, which is positive semidefinite as $p(\boldsymbol{x})$ is nonnegative. We note that numerical integration is required to evaluate the Hessian and gradients. For an $n$-dimensional system with moments up to order $r$ are matched, computing the exact gradients and Hessians requires evaluating $|\mathbb{N}_r^n|$ and $|\mathbb{N}_{2r}^n|$ terms of moments, respectively.

For nonempty $\mathcal{K}$ defined by $\boldsymbol{g}(\boldsymbol{x})$ in Assumption 1, additional linear constraints is needed in (4) to consider the ideal generated by $\boldsymbol{g}(\boldsymbol{x})$, i.e., $\mathcal{I}[\boldsymbol{g}(\boldsymbol{x})] := \{\sum_i q_i(\boldsymbol{x})g_i(\boldsymbol{x}) \mid \forall q_i(\boldsymbol{x}) \in \mathbb{R}[\boldsymbol{x}]\}$. The detailed derivation is summarized in Appendix F.

---

[6][40] considers the $1-$norm $\|p(\boldsymbol{x})\|_1 := \int_{\mathcal{K}} |p(\boldsymbol{x})|d\boldsymbol{x}$.

# 6 Main MEM-KF Algorithm

In this section, we introduce the Max Entropy Moment Kalman Filter (MEM-KF) algorithm. In the prediction steps, instead of directly marginalizing the distribution, we predict the moments via the dynamics first, and then recover the MED matching those moments. In the update steps, we formulate the observation model as MED and update the parameters via Maximum A Posterior estimation.

## 6.1 Extended Dynamical System

To generate higher-order moments of the states to better approximate the belief, we need an extended version of system (1) that exposes these moments in the state-space model. To achieve this goal, we apply $\phi_r(\cdot)$ to both sides of (1) to generate monomials of degree up to $r$ for $\boldsymbol{x}_k, \boldsymbol{w}_k$ and $\boldsymbol{v}_k$:

$$
\begin{cases} \phi_r(\boldsymbol{x}_{k+1}) = \phi_r(\boldsymbol{f}(\boldsymbol{x}_k, \boldsymbol{u}_k, \boldsymbol{w}_k)) \\ \phi_r(\boldsymbol{h}(\boldsymbol{y}_k, \boldsymbol{x}_k)) = \phi_r(\boldsymbol{v}_k) \end{cases} \Rightarrow \begin{cases} \phi_r(\boldsymbol{x}_{k+1}) = \boldsymbol{A}(\boldsymbol{w}_k, \boldsymbol{u}_k)\phi_s(\boldsymbol{x}_k) \\ \boldsymbol{C}(\boldsymbol{y}_k)\phi_q(\boldsymbol{x}) = \phi_r(\boldsymbol{v}_k) \end{cases}. \tag{6}
$$

Intuitively, (6) combines the entries of the original process and measurement models into higher-order polynomials by taking powers and products of the entries of $\boldsymbol{f}$ and $\boldsymbol{h}$. Since each polynomial of degree up to $r$ can be written as a linear combination of the canonical basis $\phi_r(\boldsymbol{x})$, we can rewrite both the extended process or the observation models as linear functions of $\phi_r(\boldsymbol{x})$. Note that as the $\phi_s(\boldsymbol{x}_k)$ on the RHS of the dynamics may have different degrees as LHS due to the nonlinearity, thus we have the order possibly $r \neq s, r \neq q$. We also note that the first term of $\phi_r(\cdot)$ is always 1 so we have the dynamics as homogeneous linear equalities. An example of the extended dynamical system is presented in Appendix G.

## 6.2 Initialization

We denote the belief or parameters after the prediction step with superscript $(\cdot)^-$. For a recursive estimator, we set the initial state as a MED $p(\boldsymbol{x}|\boldsymbol{\lambda}_0) = \exp\left(-\sum_\alpha \lambda_{0,\alpha}\boldsymbol{x}^\alpha\right)$ with coefficients $\boldsymbol{\lambda}_0$. One typical choice is the Gaussian distribution, a max-entropy distribution with constraints on the mean and covariance.

## 6.3 Prediction Steps

It is possible to formulate the prediction step as marginalizing the joint density between $\boldsymbol{x}_k$ and $\boldsymbol{x}_{k+1}$:

$$
p^-(\boldsymbol{x}_{k+1}) \propto \int_{\mathcal{K}} p(\boldsymbol{x}_{k+1}|\boldsymbol{x}_k, \boldsymbol{u}_k)p(\boldsymbol{x}_k|\boldsymbol{\lambda}_k)d\boldsymbol{x}_k. \tag{7}
$$

However, the integration on the RHS is generally intractable for arbitrary distribution. Even if $p(\boldsymbol{x}_{k+1}|\boldsymbol{x}_k, \boldsymbol{u}_k)$ and $p(\boldsymbol{x}_k|\boldsymbol{\lambda}_k)$ are MED, $p^-(\boldsymbol{x}_{k+1})$ is not guaranteed to be MED, that is, we can not claim MED is a valid conjugate prior in general.

To mitigate this issue, we instead use the extended dynamics to predict the moments for $p^-(\boldsymbol{x}_{k+1})$ and then recover the distribution. Consider the extended dynamics and take the mean value on both sides:

$$
\phi_r(\boldsymbol{x}_{k+1}) = \boldsymbol{A}(\boldsymbol{w}_k, \boldsymbol{u}_k)\phi_s(\boldsymbol{x}_k) \Rightarrow \mathbb{E}[\phi_r(\boldsymbol{x}_{k+1})] = \mathbb{E}[\boldsymbol{A}(\boldsymbol{w}_k, \boldsymbol{u}_k)\phi_s(\boldsymbol{x}_k)]. \tag{8}
$$

Under the assumption that the noise distribution is independent of the state distribution, the expectation of the cross term of $\boldsymbol{x}$ and $\boldsymbol{w}$ can be separated, and the above equation can be simplified to:

$$
\bar{\boldsymbol{x}}_{\alpha,k+1} = \boldsymbol{A}(\bar{\boldsymbol{w}}_\gamma, \boldsymbol{u}_k)\bar{\boldsymbol{x}}_{\beta,k}, \tag{9}
$$

with the moment sequence specified by $\alpha \in \mathbb{N}_r^n, \beta \in \mathbb{N}_s^n$. Given the moments sequence $\bar{\boldsymbol{x}}_{\alpha,k+1}$, we can apply (4) to recover the distribution represented by $\boldsymbol{\lambda}_{k+1}^-$, i.e., $p(\boldsymbol{x}_{k+1}|\boldsymbol{\lambda}_{k+1}^-)$.

**Remark 3.** *According to Theorem 1, $p^-(\boldsymbol{x}_{k+1})$ converges to the true distribution in norm as $r \to \infty$, which, in this limiting case, marginalizes the distribution (7) without the symbolic integration.*

## 6.4 Update Steps

In the update steps, we apply Maximum A Posterior estimation to update the belief. Consider the extended measurement model $\boldsymbol{C}(\boldsymbol{y})\phi_q(\boldsymbol{x}) = \phi_r(\boldsymbol{v})$ Given the moments of $\phi_r(\boldsymbol{v})$, i.e., $\bar{\boldsymbol{v}}_\alpha$, we model the noise distribution as MED parameterized by $\boldsymbol{\mu}$:

$$
p(\boldsymbol{v}|\boldsymbol{\mu}) = \exp\left(-\sum_\alpha \mu_\alpha \boldsymbol{v}^\alpha\right). \tag{10}
$$

We note that the noise $\boldsymbol{v}$ relates to the observation model via $\boldsymbol{v} = \boldsymbol{h}(\boldsymbol{y}, \boldsymbol{x})$. Thus, we have the likelihood function by substituting the $\boldsymbol{v}$ with $\boldsymbol{h}(\boldsymbol{y}, \boldsymbol{x})$:

$$p(\boldsymbol{y}|\boldsymbol{x}, \boldsymbol{\nu}) = \exp\left(-\textstyle\sum_\alpha \mu_\alpha \boldsymbol{h}(\boldsymbol{y}, \boldsymbol{x})^\alpha\right) = \exp\left(-\textstyle\sum_\alpha \nu_\alpha \boldsymbol{x}^\alpha\right), \tag{11}$$

where $\boldsymbol{\nu}$ is the coefficients corresponding to the canonical basis $\phi_r(\boldsymbol{x})$. With a batch of $m$ measurements, and the prior distribution $p(\boldsymbol{x}|\boldsymbol{\lambda}^-)$, the posterior distribution of $\boldsymbol{x}$ can be obtained by:

$$p(\boldsymbol{x}|\boldsymbol{\lambda}) \propto p(\boldsymbol{x}|\boldsymbol{\lambda}^-) \textstyle\prod_{i=1}^m p(\boldsymbol{y}_i|\boldsymbol{x}, \boldsymbol{\mu}) = \exp\left(-\textstyle\sum_\alpha \lambda_\alpha \boldsymbol{x}^\alpha\right), \tag{12}$$

with $\boldsymbol{\lambda}$ being the summation of the coefficients $\boldsymbol{\lambda} = \boldsymbol{\lambda}^- + \sum_i^m \boldsymbol{\nu}_i$. To predict the belief of the next step, we normalize the likelihood function via numerical integration to compute the moments.

### 6.5 Extraction of Optimal Point Estimation

Given the belief at each step of MEM-KF, we extract the optimal point estimation with the highest probability density by:

$$\max_{\boldsymbol{x} \in \mathcal{K}} \ p(\boldsymbol{x}|\boldsymbol{\lambda}) \Leftrightarrow \min_{\boldsymbol{x} \in \mathcal{K}} \ -\ln p(\boldsymbol{x}|\boldsymbol{\lambda}). \tag{13}$$

Given Assumption 1 and (3), (13) is a (POP) with the polynomial cost function being the negative log-likelihood. Thus, we instead solve the semidefinite moment relaxation of (POP) (see details in Appendix A) to extract the optimal point estimate that has the highest density value. Under the condition that the resulting SDP has a rank-1 solution, the optimal point estimate can be extracted and certified by the rank condition. We summarize MEM-KF in Algorithm 1.

**Remark 4** (KF as a subset of MEM-KF). *The KF is a special case of MEM-KF with linear process and measurement models, and the moments up to the second order are matched.*

---

**Algorithm 1** Max Entropy Moment Kalman Filter

**Require:** Initial distribution parameterized by $\boldsymbol{\lambda}_0^+$, dynamics model $\boldsymbol{x}_{k+1} = \boldsymbol{f}(\boldsymbol{x}_k, \boldsymbol{u}_k, \boldsymbol{w}_k)$, observation model $\boldsymbol{h}(\boldsymbol{y}, \boldsymbol{x}) = \boldsymbol{v}$.

```
// Obtain the observation model
```
$\boldsymbol{\mu}_\beta \xleftarrow{(4)} \bar{\boldsymbol{v}}_\beta$
**for** time $k = 1, \ldots, N$ **do**
```
    // Compute the moments at k-1
```
$\quad \bar{\boldsymbol{x}}_{\beta, k-1} \leftarrow \int_{\mathcal{K}} p(\boldsymbol{x}|\boldsymbol{\lambda}_{k-1}) \boldsymbol{x}^\beta d\boldsymbol{x}$
```
    // Propagation of moments
```
$\quad \bar{\boldsymbol{x}}_{\alpha, k} \leftarrow \boldsymbol{A}(\bar{\boldsymbol{w}}_\alpha, \boldsymbol{u}_{k-1}) \bar{\boldsymbol{x}}_{\beta, k-1}$
```
    // Reconstruct the distribution
```
$\quad \boldsymbol{\lambda}_k^- \xleftarrow{(4)} \bar{\boldsymbol{x}}_{\alpha, k}$
```
    // Update with measurements
```
$\quad \boldsymbol{\nu}_k \xleftarrow{(11)} \boldsymbol{y}_k, \boldsymbol{\mu}_\beta \quad \boldsymbol{\lambda}_k \leftarrow \boldsymbol{\lambda}_k^- + \boldsymbol{\nu}_k$
```
    // Normalize the distribution
```
$\quad \lambda_{k,\alpha_0} \leftarrow \lambda_{k,\alpha_0} - \ln \int_{\mathcal{K}} p(\boldsymbol{x}|\boldsymbol{\lambda}_k) d\boldsymbol{x}$
```
    // Optimal point estimation
```
$\quad \boldsymbol{x}_k^*, \boldsymbol{X}_k^* \xleftarrow{(\text{SDP})} \min_{\boldsymbol{x} \in \mathcal{K}} -\ln p(\boldsymbol{x}|\boldsymbol{\lambda}_k)$
**end for**

---

## 7 Numerical Experiments

### 7.1 Asymptotic Property of MED

We first showcase the asymptotic property of MED in approximating complicated distributions. We apply a line search method to implement (4) with the initial step size determined by exact first-order gradients (5) and the Hessian approximated by the BFGS algorithm. Although Newton's method is introduced in [42], we find obtaining the exact Hessian matrix extremely expensive for high-order systems. We adopt SPOTLESS [43] to process the polynomials and an adaptive Gaussian quadrature method [44] for numerical integrations. We apply the MOSEK [45] to solve the (SDP) to extract the point estimations. More results on nonempty $\mathcal{K}$ like SE(2) can be found in the Appendix. H.2.

In the first case, we consider a distribution on $\mathbb{R}^2$ generated by the following sampling strategy:

$$\boldsymbol{v} = \begin{bmatrix} \sin w + \epsilon_1 \\ \cos \pi w + \epsilon_2 \end{bmatrix} \tag{14}$$

where $w \sim U(0, \pi)$ is a uniform distribution between 0 and $\pi$, and $\epsilon_1, \epsilon_2 \sim \mathcal{N}(0, 0.01)$ is Gaussian noise. We draw $N = 1e5$ samples from the distribution to compute mean values, an unbiased estimation of the underlying moments, i.e., $\hat{\boldsymbol{v}}_\alpha = \frac{1}{N} \sum_{k=1}^N \boldsymbol{v}_k^\alpha, \alpha \in \mathbb{N}_r^n$.

Then we recover the distribution on $\mathbb{R}^2$ as $p(\boldsymbol{x}|\boldsymbol{\lambda})$. The samples and MED with moment up to order $r$ are shown in Figure 2. We observe that as more moments are matched, the distribution asymptotically approaches the sample distribution. The result is expected according to Theorem 1. When $r = 2$, the MED is the Gaussian distribution. Due to the nonlinear transformation, the samples are concentrated on the curve with higher curvatures, which starts emerging in $p(\boldsymbol{x}|\boldsymbol{\lambda})$ when $r \geq 4$. We can further verify that the result of $r = 12$ is sufficiently good by comparing it with the heat map of the samples.

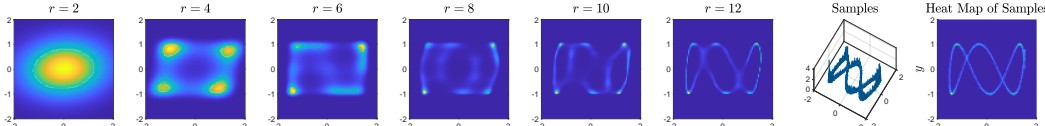

Figure 2: The MED with moment matched up to order $r$. The heat map and samples are shown on the RHS of the plot. MED asymptotically approaches the sample distribution as the order increases.

Table 1: Covariance of rank-1 point estimation with $N$ accumulated samples using noise model (14) and (15) with moments up to order $r$ matched. The normalized log-likelihood of the belief, i.e., $\ln p(\boldsymbol{x})$, is shown in $(\cdot)$ for small $N$. We note that increasing $r$ consistently results in higher density.

| | $N =$ 
 $r =$ | 1 | 2 | 3 | 5 | 10 | 20 | 50 | 100 |
|---|---|---|---|---|---|---|---|---|---|
| (14) | 2 (BLUE) | 1.58 (−1.54) | 0.805 (−0.844) | 0.539 (−0.439) | 0.326 | 0.163 | 0.0784 | 0.0319 | 0.0161 |
| | 4 | 3.38 (−0.840) | 1.58 (0.034) | 0.771 (0.665) | 0.187 | 0.0303 | 0.0101 | $3.47e{-}3$ | $1.71e{-}3$ |
| | 6 | 4.48 (−0.161) | 1.80 (0.885) | 0.696 (1.65) | 0.159 | 0.0210 | $3.54e{-}3$ | $8.94e{-}4$ | $4.27e{-}4$ |
| | 6 | 4.52 (0.892) | 1.83 (2.03) | 0.661 (2.81) | 0.131 | $7.09e{-}3$ | $9.30e{-}4$ | $2.72e{-}4$ | $1.34e{-}4$ |
| | 10 | 4.04 (1.26) | 1.83 (3.04) | 0.628 (3.96) | 0.0596 | $1.06e{-}3$ | $3.08e{-}4$ | $1.07e{-}4$ | $4.69e{-}5$ |
| | 12 | 2.41 (1.72) | 1.72 (4.25) | 0.367 (5.92) | 0.0230 | $3.78e{-}4$ | $1.75e{-}4$ | $7.12e{-}5$ | $3.27e{-}5$ |
| (15) | 2 (BLUE) | 2.07 (−1.88) | 1.01 (−1.18) | 0.706 (−0.779) | 0.420 | 0.208 | 0.0989 | 0.0407 | 0.0199 |
| | 4 | 4.47 (0.073) | 2.05 (1.42) | 1.14 (2.22) | 0.272 | 0.0128 | $4.59e{-}3$ | $1.90e{-}3$ | $9.36e{-}4$ |

## 7.2 Update Steps

Now we verify the update step of MEM-KF. We consider a basic linear observation model $\boldsymbol{y} = \boldsymbol{x} + \boldsymbol{v} \in \mathbb{R}^2$, where the moments of $\boldsymbol{v}$ are known in advance. Then, we consider the non-Gaussian noise generated by the following nonlinear sampling strategy:

$$\boldsymbol{v} = \begin{bmatrix} 2q_1 - 1 + \epsilon_1 \\ 2q_2 - 1 + \epsilon_2 \end{bmatrix}, q_1, q_2 \sim \text{Bernoulli}(0.5), \epsilon_1, \epsilon_2 \sim \mathcal{N}(0, 0.2\boldsymbol{I}). \tag{15}$$

We compare the proposed method with the Best Linear Unbiased Estimator (BLUE)[46], which is the minimum variance estimator using only mean and covariance. We compare the asymptotic property of the update steps with accumulated measurements. We accumulate $N$ measurements to obtain the negative log-likelihood, i.e., $-\ln p(\boldsymbol{x}|\boldsymbol{y}_1, \boldsymbol{y}_2, \cdots, \boldsymbol{y}_N) = \text{Scalar} + \sum_{k=1}^{N} \sum_\alpha \mu_\alpha \boldsymbol{h}(\boldsymbol{y}_k, \boldsymbol{x})^\alpha$.

The normalized belief distribution is illustrated in Figure 3. After receiving the first measurement, we notice that the belief computed by $r = 4$ retains the multi-modality caused by the noise distribution and exhibits four different peaks. After more measurements are incorporated, the belief converges to the true state with dramatically smaller covariance. On the other hand, the BLUE ($r = 2$) provides a unimodal (Gaussian) approximation of the belief and even with more data, it still exhibits a relatively large covariance around the true state. The interested reader can also find results with noise (14) are shown in Figure 6 in the Appendix. H.1 which exhibit similar patterns.

We further sample 1000 trials in each noise with different $N$ to compute the empirical variance of the point estimation extracted by (SDP). The results are presented in Table 1. We observe that with sufficiently many measurements accumulated ($N \geq 5$), increasing $r$ consistently lowers the covariance and outperforms BLUE dramatically. With small $N$, the multi-modality of the belief introduces additional bias when computing the point estimation that makes the covariance higher than BLUE. However, we note that the point estimation is not a full belief representation and can not represent the multi-modality. The log-likelihood of the normalized density for the cases with $N \leq 3$ is computed to show that increasing $r$ can provide point estimations with higher likelihood.

## 7.3 Localization with Unknown Data Association on $\text{SE}(2)$

Now we consider a localization problem where a robot with pose:

$$\boldsymbol{Z} := \begin{bmatrix} \boldsymbol{R} & \boldsymbol{p} \\ 0 & 1 \end{bmatrix} \in \text{SE}(2), \text{ including rotation } \boldsymbol{R} := \begin{bmatrix} c & -s \\ s & c \end{bmatrix} \text{ and position } \boldsymbol{p} := \begin{bmatrix} p_x \\ p_y \end{bmatrix}$$

moves in the plane while collecting range-and-bearing measurements.[7] The problem is modeled by the following dynamical system:

$$\begin{cases} \boldsymbol{Z}_{k+1} = \boldsymbol{Z}_k \boldsymbol{U}_k \boldsymbol{W}_k \in \text{SE}(2) \\ \boldsymbol{y}_k = \boldsymbol{R}_k^\mathsf{T}(\boldsymbol{L}_{ik} + \boldsymbol{v} - \boldsymbol{p}_k) \in \mathbb{R}^2 \end{cases}, \tag{16}$$

---

[7]Note that the unknown rotation has to satisfy the equality constraint $g(\boldsymbol{x}) = c^2 + s^2 - 1 = 0$, due to the structure of the Special Orthogonal group $\text{SO}(2)$.

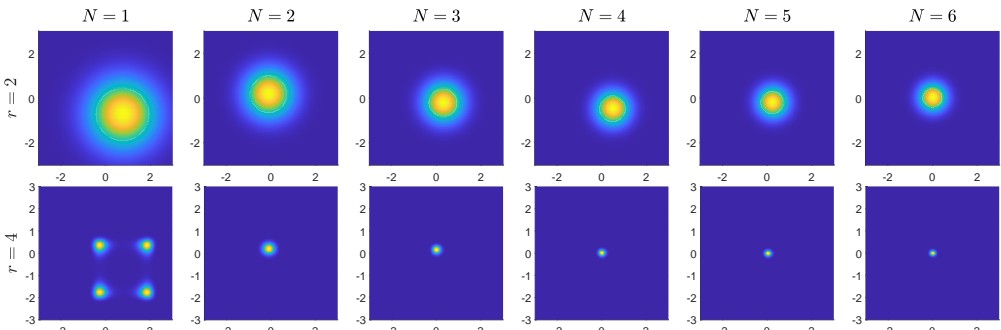

Figure 3: Applying the update step of MEM-KF to the linear system corrupted by noise (15). With larger $r$, the posterior distribution converges faster to the truth states. The case for noise (14) is presented in the Appendix.

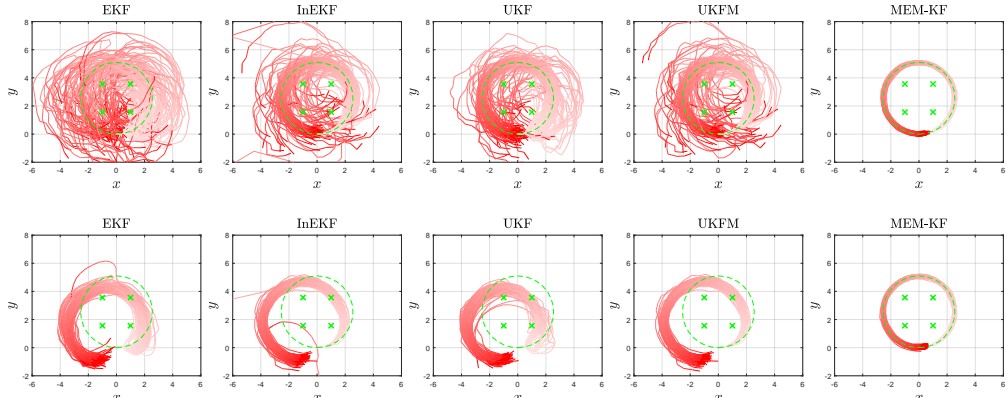

Figure 4: Localization with unknown data association. In the first row, the landmark association follows the same distribution in (17). In the second row, the measurement comes more from the landmarks at the lower left corner. The performance of baselines degrades due to incorrect data associations, while MEM-KF can consistently track the ground truth.

where the pose $\boldsymbol{Z}$, input $\boldsymbol{U}$, and noise $\boldsymbol{W}$ all belong to SE(2). $\boldsymbol{U}$ models the wheel odometry of a differential-drive wheel robot, and $\boldsymbol{W}$ can be generated by the associated noise distribution in the Lie algebra [47]. Due to the SO(2) group constraints on $\boldsymbol{R}$, the system is nonlinear. We denote with $\boldsymbol{L}_i \in \mathbb{R}^2$ the (known) position of landmark $i \in \mathcal{I}$ and with $\boldsymbol{L}_{ik}$ the position of the landmark observed at time $k$, while $\boldsymbol{v}$ is the measurement noise.

In our case study, we consider the robot receiving observations from four landmarks with unknown data associations. Due to the unknown data associations, for given $\boldsymbol{y}_k$, we do not know which landmark is actually measured, so the index $ik$ is unknown. The work [48] decides the data association between observations and landmarks by taking the $ik$ as variables. For a filtering problem, [48] is equivalent to choosing the most likely association and leads to an intractable estimation problem. To mitigate this issue, we assume each measurement to one of the landmarks with equal probability $p_1 = p_2 = \cdots = p_N = \frac{1}{N}$. Thus, such "sensor model" with the max entropy association strategy can be formulated as

$$\boldsymbol{R}_k \boldsymbol{y}_k + \boldsymbol{p}_k = \boldsymbol{v}_L := \boldsymbol{L}_{ik} + \boldsymbol{v}, \tag{17}$$

where $p(\boldsymbol{L}_{ik} = \boldsymbol{L}_i) = \frac{1}{N}, \forall i \in \mathcal{I}$ and $\boldsymbol{v} \in \mathcal{N}(0, \boldsymbol{\Sigma})$. The RHS can be modeled as a single noise term $\boldsymbol{v}_L$ that can be approximated by MED following the same procedure for (15). We consider $r = 4$ in our case, which is sufficiently good to represent the four peaks caused by the landmarks. For $r = 2$, representing the four landmarks as a Gaussian distribution with extremely large covariance leads to little innovation in the update steps.

We present the result of the proposed method and compare it with the baselines, including the EKF, UKF, and their variants on SE(2) manifolds, i.e., the InEKF [16] and UKFM [49]. We consider two scenarios, such that 1) the underlying association distribution is identical to the max entropy association, i.e., the measurement comes from four landmarks with the equal probability, and 2) the robot receives more measurements from the landmark on the lower-left corner. The results of the two

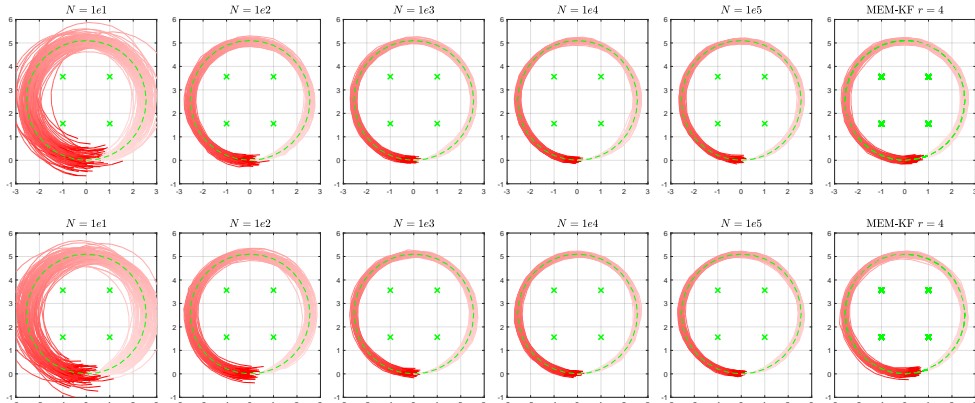

Figure 5: Comparison of particle filter with $N$ particles with MEM-KF. In the first row, the landmark association follows the same distribution in (17). In the second row, the measurement comes more from the landmarks at the lower left corner. We see that the particle filter can better track the trajectories with more particles. We find that both the particle filter with a sufficient number of particles and MEM-KF can estimate the trajectories with similar quality. The statistics are presented in Figure 8.

cases are shown in Figure 4. We can see that MEM-KF consistently outperforms the baselines in both scenarios. The baselines in 2) have a clear bias in the trajectory, due to the fact that they greedily choose the most likely landmark as the originator of a measurement.

The MEM-KF is also compared with the particle filter in Figure 5. When sufficiently many particles are added, both methods can successfully estimate the trajectories with similar estimation error. However, we note that MEM-KF is deterministic and provides a smooth belief representation that is analytical and convenient for the extraction of the globally optimal point estimation via convex optimization. Such a property is not possible for the particle filter that is based on random sampling. More statistics about the comparison are presented in the Appendix. H.3.

## 8 Conclusions

In this work, we presented the Max Entropy Moment Kalman Filter (MEM-KF) for optimal state estimation of polynomial systems corrupted by arbitrary noise. By the Moment-Constrained Max-Entropy Distribution (MED), we can asymptotically approximate almost any belief and noise distributions given an increasing number of moments of the underlying distributions. We then leverage the system dynamics to predict future moments to overcome the difficulties in the marginalization steps in the prediction steps. The update step is formulated as a Maximum A Posteriori estimation problem with measurements modeled as MED. All the steps in MEM-KF, as well as the extraction of the optimal point estimation, can be exactly solved by convex optimization. We showcase the MEM-KF in challenging robotics tasks, such as localization with unknown data association.

**Limitations and Future Work** The proposed method relies on numerical integration, which is computationally intensive for real-time deployment. The Gaussian-quadrature applied to evaluate the gradients (5) in (4) still scales exponentially w.r.t the dimension of the systems. Given this reason, the average time consumption for the localization case in the update is 6.87 seconds and 34.51 seconds in recovering MED from moments. Future work will consider GPU-based numerical integration that scales better for high-dimensional systems and real-time deployment. Possible implementation can be [50], which includes a GPU version of the deterministic integration methods in [51]. To further improve the scalability of MEM-KF, we can leverage the sparsity of polynomials by considering only a partial amount of moments [52].

**Societal Impact** This paper provides an optimal filtering framework that can asymptotically approximate any moment-determinant belief for non-Gaussian polynomial systems with smooth functions. This paper pushes the boundary of reliable and certifiable state estimations, especially for robotics.

## Acknowledgment

This work was partially funded by the ONR RAPID program and Carlone's NSF CAREER. Maani Ghaffari and Ram Vasudevan are sponsored by AFOSR grant FA9550-23-1-0400 (MURI).

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

## A Polynomial Optimization and Semidefinite Relaxations

In general, POPs are hard non-convex problems. In order to obtain a convex relaxation, we first rewrite the POP as a function of the moment matrix. The key idea here is that we can write a polynomial $c(\boldsymbol{x})$ of degree up to $2r$ as a linear function of the moment matrix $\boldsymbol{M}_r(\boldsymbol{x})$, which contains all the monomials of degree up to $2r$:

$$c(\boldsymbol{x}) = \operatorname{tr}(\boldsymbol{C}\boldsymbol{M}_r(\boldsymbol{x})) := \langle \boldsymbol{C}, \boldsymbol{M}_r(\boldsymbol{x}) \rangle, \tag{18}$$

where we observe that $\boldsymbol{M}_r(\boldsymbol{x})$, by construction, is a positive semidefinite rank-1 matrix. We then relax the problem by dropping the rank-one constraint on the moment matrix $\boldsymbol{M}_r(\boldsymbol{x})$ while enforcing:

$$\boldsymbol{X} \succeq 0, \boldsymbol{X}_{1,1} = 1. \qquad \text{(Semidefinite relaxation)}$$

Since the moment matrix $\boldsymbol{M}_r(x)$ contains multiple repeated entries, we further add *moment constraints* that enforce these repeated entries to be identical:

$$\langle \boldsymbol{B}_i^\perp, \boldsymbol{X} \rangle = 0. \qquad \text{(Moment constraints)}$$

Finally, we enforce the equality constraints, using *localizing constraints* in the form:

$$\langle \boldsymbol{G}_j, \boldsymbol{X} \rangle = 0. \qquad \text{(Localizing constraints)}$$

These steps lead to the standard semidefinite moment relaxation of (POP), see [22, 53].

**Definition 3** (Moment Relaxation of (POP) [22, 53])**.**

$$\hat{\rho} := \inf_{\boldsymbol{X}} \langle \boldsymbol{C}, \boldsymbol{X} \rangle$$
$$\text{s.t.} \quad \boldsymbol{X} \succeq 0, \boldsymbol{X}_{1,1} = 1 \tag{SDP}$$
$$\langle \boldsymbol{B}_i^\perp, \boldsymbol{X} \rangle = 0, \langle \boldsymbol{G}_j, \boldsymbol{X} \rangle = 0 \quad \forall i, j.$$

If a rank-1 solution is obtained from (SDP), its solution $\hat{\boldsymbol{X}}$ is a valid moment matrix. In such a case, we can extract $\hat{\boldsymbol{X}} = \phi_r(\hat{\boldsymbol{x}})\phi_r^\mathsf{T}(\hat{\boldsymbol{x}})$ from the first column of $\hat{\boldsymbol{X}}$.

## B Auxiliary Matrices

This appendix formally defines the localizing matrices $\{\boldsymbol{B}_\gamma\}_\gamma$ [53] that locate the monomials in a moment matrix and its orthogonal complement $\{\boldsymbol{B}_{\gamma,\alpha}^\perp\}_{\gamma,\alpha}$. These matrices are used in the definition of the moment constraints for the moment relaxations for (POP).

We first introduce the matrix $\boldsymbol{B}_\gamma$ which retrieves the monomial $\boldsymbol{x}^\gamma$ from the moment matrix $\boldsymbol{M}_r(\boldsymbol{x})$, i.e., $\langle \boldsymbol{B}_\gamma, \boldsymbol{M}_r(\boldsymbol{x}) \rangle = \boldsymbol{x}^\gamma$:

$$\boldsymbol{B}_\gamma := \frac{1}{C_\gamma} \sum_{\gamma = \alpha + \beta} \boldsymbol{e}_\alpha \boldsymbol{e}_\beta^\mathsf{T}, \quad C_\gamma := |\{(\alpha, \beta) | \gamma = \alpha + \beta\}|, \tag{19}$$

where $\boldsymbol{e}_\alpha \in \mathbb{R}^{s(r,n) \times 1}$ is the canonical basis corresponding to the $n$-tuples $\alpha \in \mathbb{N}_r^n$ and $C_\gamma$ is a normalizing factor. We note that the moment matrix can be written as:

$$\boldsymbol{M}_r(\boldsymbol{x}) = \sum_\gamma C_\gamma \boldsymbol{B}_\gamma \boldsymbol{x}^\gamma, \quad \gamma \in \mathbb{N}_{2r}^n. \tag{20}$$

Then we introduce $\boldsymbol{B}_{\gamma,\alpha}^\perp$, which is used to enforce that repeated entries (corresponding to the same monomial $\boldsymbol{x}^\gamma$) in the moment matrix are identical. We first define the symmetric matrix, for any given $\alpha, \beta \in \mathbb{N}_r^n$:

$$\boldsymbol{E}_{\alpha,\beta} = \boldsymbol{e}_\alpha \boldsymbol{e}_\beta^\mathsf{T} + \boldsymbol{e}_\beta \boldsymbol{e}_\alpha^\mathsf{T}. \tag{21}$$

Then we construct $\boldsymbol{B}_{\gamma,\alpha}^\perp$, for some given $\gamma \in \mathbb{N}_{2r}^n$ and for $\alpha \in \mathbb{N}_r^n$ (with $\alpha < \gamma$) as:

$$\boldsymbol{B}_{\gamma,\alpha}^\perp = \boldsymbol{E}_{\bar{\alpha},\bar{\beta}} - \boldsymbol{E}_{\alpha,\beta}, \tag{22}$$

where $\beta$ is such that $\alpha + \beta = \gamma$ and $\beta \leq \alpha$, and where $\bar{\alpha}, \bar{\beta} \in \mathbb{N}_r^n$ are chosen such that $\bar{\alpha} + \bar{\beta} = \gamma$ and $\bar{\alpha}$ is the smallest vector (in the lexicographic sense) that satisfies $\bar{\alpha} + \bar{\beta} = \gamma$. An example of matrices $\boldsymbol{B}_\gamma$ and $\boldsymbol{B}_{\gamma,\alpha}^\perp$ for a simple problem is given in Appendix C.

## C Examples of Moment Matrix and Constraints

For $r = 2, n = 2$, we have the moment matrix:

$$M_2(x) = \begin{bmatrix} 1 & x_1 & x_2 & x_1^2 & x_1x_2 & x_2^2 \\ x_1 & x_1^2 & x_1x_2 & x_1^3 & x_1^2x_2 & x_1x_2^2 \\ x_2 & x_1x_2 & x_2^2 & x_1^2x_2 & x_1x_2^2 & x_2^3 \\ x_1^2 & x_1^3 & x_1^2x_2 & x_1^4 & x_1^3x_2 & x_1^2x_2^2 \\ x_1x_2 & x_1^2x_2 & x_1x_2^2 & x_1^3x_2 & x_1^2x_2^2 & x_1x_2^3 \\ x_2^2 & x_1x_2^2 & x_2^3 & x_1^2x_2^2 & x_1x_2^3 & x_2^4 \end{bmatrix}.$$

For $\gamma = [2, 0]$, which corresponds to $x_1^2$, we have:

$$B_\gamma = \frac{1}{C_\gamma} \begin{bmatrix} 0 & 0 & 0 & 1 & 0 & 0 \\ 0 & 1 & 0 & 0 & 0 & 0 \\ 0 & 0 & 0 & 0 & 0 & 0 \\ 1 & 0 & 0 & 0 & 0 & 0 \\ 0 & 0 & 0 & 0 & 0 & 0 \\ 0 & 0 & 0 & 0 & 0 & 0 \end{bmatrix}, C_\gamma = 3. \tag{23}$$

Now let us compute $B_{\gamma,\alpha}^\perp$ for $\alpha = [1, 0] \leq \gamma$. We note that $\beta = \gamma - \alpha = [1, 0]$ and we can choose $\bar{\alpha} = [0, 0]$ and $\bar{\beta} = [2, 0]$. The corresponding vectors in the canonical basis are:

$$\begin{aligned} e_\alpha &= [0 \quad 1 \quad 0 \quad 0 \quad 0 \quad 0], e_\beta &= [0 \quad 1 \quad 0 \quad 0 \quad 0 \quad 0], \\ e_{\bar{\alpha}} &= [1 \quad 0 \quad 0 \quad 0 \quad 0 \quad 0], e_{\bar{\beta}} &= [0 \quad 0 \quad 0 \quad 1 \quad 0 \quad 0], \end{aligned} \tag{24}$$

from which we obtain:

$$\begin{aligned} B_{\gamma,\alpha}^\perp &= E_{\bar{\alpha},\bar{\beta}} - E_{\alpha,\beta} = e_{\bar{\alpha}}e_{\bar{\beta}}^\mathsf{T} + e_{\bar{\beta}}e_{\bar{\alpha}}^\mathsf{T} - (e_\alpha e_\beta^\mathsf{T} + e_\beta e_\alpha^\mathsf{T}) \\ &= \begin{bmatrix} 0 & 0 & 0 & +1 & 0 & 0 \\ 0 & -2 & 0 & 0 & 0 & 0 \\ 0 & 0 & 0 & 0 & 0 & 0 \\ +1 & 0 & 0 & 0 & 0 & 0 \\ 0 & 0 & 0 & 0 & 0 & 0 \\ 0 & 0 & 0 & 0 & 0 & 0 \end{bmatrix}. \end{aligned} \tag{25}$$

By adding the constraint:

$$\langle B_{\gamma,\alpha}^\perp, X \rangle = 0, \quad X \succeq 0, \tag{26}$$

we enforce that the entries corresponding to $x_1^2$ are identical.

## D Derivation of MED

Consider the optimization $(P_r)$, given the multiplier $\lambda \in \mathbb{R}^{|\mathbb{N}_r^n|}$, we have the Lagrangian:

$$\mathcal{L}(p(x), \lambda) = \int_\mathcal{K} p(x) \ln p(x) dx + \sum_\alpha \lambda_\alpha (\int_\mathcal{K} x^\alpha p(x) dx - \bar{x}_\alpha). \tag{27}$$

For fixed $p(x)$, the dual problem to $(P_r)$ becomes:

$$\max_\lambda \min_{p(x)} \quad \mathcal{L}(p(x), \lambda) \tag{28}$$

Via taking the variation of $p(x)$ on $\mathcal{L}(p(x), \lambda)$, we have the first-order optimality condition:

$$\nabla_{p(x)} \mathcal{L}(p(x), \lambda) = p(x) \frac{1}{p(x)} + \ln p(x) + \sum_\alpha \lambda_\alpha x^\alpha, \tag{29}$$

where $\nabla$ is the Fréchet derivative in a normed linear space defined on all possible distributions. For simplicity, we merge the constant 1 produced by the entropy term and let the (29) equal to zero. Then we have the stationary point (3):

$$p(x) = \exp\left(-\sum_\alpha \lambda_\alpha x^\alpha\right).$$

# E  Derivation of Thermal Dynamics Potential

Now we introduce the thermal dynamics potential term that we can minimize to obtain the coefficients of (3). We mainly refer to the result in [42] to introduce the thermal dynamics potential and the equivalent potential-like term for the optimization. Consider the integrand:

$$\varphi(\boldsymbol{\lambda}) = \int_{\mathcal{K}} \exp\left(-\sum_{\alpha \in \mathbb{N}_r^n} \lambda_\alpha \boldsymbol{x}^\alpha\right) d\boldsymbol{x}, \tag{30}$$

the normalized version of (3) can be expressed as:

$$p^*(\boldsymbol{x}) = \frac{\exp\left(-\sum_\alpha \lambda_\alpha \boldsymbol{x}^\alpha\right)}{\varphi(\boldsymbol{\lambda})}. \tag{31}$$

By substituting the $p^*(\boldsymbol{x})$ to the inner minimization of $\min_{p(\boldsymbol{x})} \mathcal{L}(p(\boldsymbol{x}), \boldsymbol{\lambda})$,

we have the Legendre-Fenchel dual as

$$
\begin{aligned}
&\mathcal{L}(p^*(\boldsymbol{x}), \boldsymbol{\lambda}) \\
&= \int_{\mathcal{K}} p^*(\boldsymbol{x}) \ln p^*(\boldsymbol{x}) d\boldsymbol{x} + \sum_\alpha \lambda_\alpha \left(\int_{\mathcal{K}} \boldsymbol{x}^\alpha p^*(\boldsymbol{x}) d\boldsymbol{x} - \bar{\boldsymbol{x}}_\alpha\right) \\
&= \int_{\mathcal{K}} p^*(\boldsymbol{x}) \ln \frac{p(\boldsymbol{x})}{\varphi(\boldsymbol{\lambda})} d\boldsymbol{x} + \sum_\alpha \lambda_\alpha \left(\int_{\mathcal{K}} \boldsymbol{x}^\alpha p^*(\boldsymbol{x}) d\boldsymbol{x} - \bar{\boldsymbol{x}}_\alpha\right) \\
&= \int_{\mathcal{K}} p^*(\boldsymbol{x}) \ln p(\boldsymbol{x}) d\boldsymbol{x} + \sum_\alpha \lambda_\alpha \int_{\mathcal{K}} \boldsymbol{x}^\alpha p^*(\boldsymbol{x}) d\boldsymbol{x} - \int_{\mathcal{K}} p^*(\boldsymbol{x}) \ln \varphi(\boldsymbol{\lambda}) d\boldsymbol{x} - \sum_\alpha \lambda_\alpha \bar{\boldsymbol{x}}_\alpha \\
&= -\int_{\mathcal{K}} p^*(\boldsymbol{x}) \ln \varphi(\boldsymbol{\lambda}) d\boldsymbol{x} - \sum_\alpha \lambda_\alpha \bar{\boldsymbol{x}}_\alpha \\
&= -\ln \varphi(\boldsymbol{\lambda}) - \sum_\alpha \lambda_\alpha \bar{\boldsymbol{x}}_\alpha =: -\Gamma(\boldsymbol{\lambda})
\end{aligned} \tag{32}
$$

We note that the coefficients $\lambda_{\alpha_0}$ corresponding to the normalization constraints can be extracted from $\varphi(\boldsymbol{\lambda})$ and we have:

$$
\begin{aligned}
\Gamma(\boldsymbol{\lambda}) &= \ln \int_{\mathcal{K}} \exp\left(-\sum_{\alpha \in \mathbb{N}_r^n/\{\alpha_0\}} \lambda_\alpha \boldsymbol{x}^\alpha\right) \exp(-\lambda_0) d\boldsymbol{x} + \left(\lambda_0 + \sum_{\alpha \in \mathbb{N}_r^n/\{\alpha_0\}} \lambda_\alpha \bar{\boldsymbol{x}}_\alpha\right) \\
&= \ln \int_{\mathcal{K}} \exp\left(-\sum_{\alpha \in \mathbb{N}_r^n/\{\alpha_0\} d\boldsymbol{x}} \lambda_\alpha \boldsymbol{x}^\alpha\right) d\boldsymbol{x} + \sum_{\alpha \in \mathbb{N}_r^n/\{\alpha_0\}} \lambda_\alpha \bar{\boldsymbol{x}}_\alpha
\end{aligned} \tag{33}
$$

Given the fact that $p(\boldsymbol{x})$ is normalized at the stationary point:

$$\int_{\mathcal{K}} \exp\left(-\sum_{\alpha \in \mathbb{N}_r^n/\{\alpha_0\}} \lambda_\alpha \boldsymbol{x}^\alpha\right) \exp(-\lambda_0) d\boldsymbol{x} = 1, \tag{34}$$

we have the un-normalized thermal dynamics potential:

$$\Delta(\boldsymbol{\lambda}) = \left(\int_{\mathcal{K}} \exp\left(-\sum_{\alpha \in \mathbb{N}_r^n} \lambda_\alpha \boldsymbol{x}^\alpha\right) d\boldsymbol{x} - 1\right) + \sum_{\alpha \in \mathbb{N}_r^n} \lambda_\alpha \bar{\boldsymbol{x}}_\alpha. \tag{35}$$

The fact that $\Delta(\boldsymbol{\lambda}) = \Gamma(\boldsymbol{\lambda})$ can be verified by extract the term $\bar{\boldsymbol{x}}_{\alpha_0} = 1$ and $\lambda_0 = \ln \int_{\mathcal{K}} \exp\left(-\sum_{\alpha \in \mathbb{N}_r^n/\{\alpha_0\} d\boldsymbol{x}} \lambda_\alpha \boldsymbol{x}^\alpha\right) d\boldsymbol{x}$.

# F  Recover Distribution on Quotient Ring

Now, we consider nonempty $\mathcal{K}$ that introduces an ideal structure to the systems, which is common in robotics. Consider the ideal generated by the polynomial equality constraints $\boldsymbol{g}(\boldsymbol{x})$:

$$\mathcal{I}[\boldsymbol{g}(\boldsymbol{x})] := \left\{\sum_i q_i(\boldsymbol{x}) g_i(\boldsymbol{x}) \mid \forall q_i(\boldsymbol{x}) \in \mathbb{R}[\boldsymbol{x}]\right\}. \tag{36}$$

The integration of any term in $\mathcal{I}[\boldsymbol{g}(\boldsymbol{x})]$ will be zero, i.e,

$$\int_{\mathcal{K}} p(\boldsymbol{x}) \sum_i q_i(\boldsymbol{x}) g_i(\boldsymbol{x}) d\boldsymbol{x} = 0, \forall \alpha, \tag{37}$$

which can be checked by the fact that $g_i(\boldsymbol{x}) \equiv 0, \forall \boldsymbol{x} \in \mathcal{K}$. Thus, we need to incorporate these constraints when recovering $\boldsymbol{\lambda}$ to avoid degenerate search direction or reducing the systems to the quotient ring $\mathbb{R}[\boldsymbol{x}]/\mathcal{I}[\boldsymbol{g}(\boldsymbol{x})]$.

Now, we proceed to construct the ideal from the canonical basis $\boldsymbol{\phi}_r(\boldsymbol{x})$ by linear algebra techniques. Consider the elements of $\mathbb{R}[\boldsymbol{x}]/\mathcal{I}[\boldsymbol{g}(\boldsymbol{x})]$ with finite degree $r$; each element can be expressed as a linear combination of the terms in the set:

$$\mathcal{B} = \{\boldsymbol{x}^\alpha g_i(\boldsymbol{x})| \deg \boldsymbol{x}^\alpha g_i(\boldsymbol{x}) \le r, \forall i, \forall \alpha\}. \tag{38}$$

Let the vector of all elements in $\mathcal{B}$ as $\boldsymbol{b}(\boldsymbol{x})$, we can express $\boldsymbol{b}(\boldsymbol{x})$ as linear combination of $\boldsymbol{\phi}_r(\boldsymbol{x})$, i.e,

$$\boldsymbol{b}(\boldsymbol{x}) = \boldsymbol{B}\boldsymbol{\phi}_r(\boldsymbol{x}). \tag{39}$$

Via QR decomposition of $\boldsymbol{B}^\mathsf{T}$, i.e,

$$\boldsymbol{B}^\mathsf{T} = [\boldsymbol{Q} \quad \boldsymbol{Q}_\perp] \begin{bmatrix} \boldsymbol{R} \\ \boldsymbol{0} \end{bmatrix}, \tag{40}$$

we can extract the null space of $\boldsymbol{B}$, i.e, the span of column space of $\boldsymbol{Q}_\perp$, to formulate the basis of the quotient ring. The $\boldsymbol{R}$ is an upper diagonal matrix. Then, we conclude that

$$\boldsymbol{q}(\boldsymbol{x}) = \boldsymbol{Q}_\perp^\mathsf{T} \boldsymbol{\phi}_r(\boldsymbol{x}) \tag{41}$$

is a basis for the quotient ring. In this case, the max entropy distribution can be expressed in terms of $\boldsymbol{q}(\boldsymbol{x})$:

$$p(\boldsymbol{x}) = \exp\left(-\boldsymbol{\lambda}_q^\mathsf{T} \boldsymbol{q}(\boldsymbol{x})\right), \tag{42}$$

and then we can minimize the thermal dynamics potential on the quotient ring:

$$\Delta(\boldsymbol{\lambda}_q) = \int_{\mathcal{K}} \exp\left(-\boldsymbol{\lambda}_q^\mathsf{T} \boldsymbol{q}(\boldsymbol{x})\right) + \sum_i \boldsymbol{\lambda}_q \bar{\boldsymbol{q}}. \tag{43}$$

By the definition of $\boldsymbol{\lambda}_q$ and $\boldsymbol{Q}_\perp$, we have the equivalence relation of $\boldsymbol{\lambda}$:

$$\boldsymbol{\lambda} - \boldsymbol{Q}_\perp \boldsymbol{\lambda}_q \in \mathrm{colsp}(\boldsymbol{Q}). \tag{44}$$

Thus, we can alternatively perform constrained optimization with moments on $\mathbb{R}[\boldsymbol{x}]$ by incorporating equality constraints to avoid the degenerate search directions:

$$\begin{aligned} \min_{\boldsymbol{\lambda}} \quad & \Delta(\boldsymbol{\lambda}) \\ \text{s.t.} \quad & \boldsymbol{Q}^\mathsf{T} \boldsymbol{\lambda} = 0. \end{aligned} \tag{45}$$

As the constraints are linear, the programming remains convex. When $\boldsymbol{\lambda}$ is fixed, the gradients and Hessian can be computed by numerical integration and used to determine the optimal search direction for the optimal coefficients. As the convex optimization is not agnostic to the initial guess, we can initialize $\boldsymbol{\lambda}$ by an MED with finite integrals over $\mathcal{K}$, for example, the Gaussian distribution.

# G   Example of Extended Polynomial System and Its Affine Form

We provide an example of the extended polynomial system and transform it to the affine form as in (6). Consider a linear measurement model with state $\boldsymbol{x} \in \mathbb{R}^2$, measurements $\boldsymbol{y}$, and measurement noise $\boldsymbol{w}$:

$$\begin{aligned} y_1 - x_1 = w_1, \\ y_2 - x_2 = w_2. \end{aligned} \tag{46}$$

The extended system at $r = 2$ is:

$$\boldsymbol{\phi}_r(\boldsymbol{y} - \boldsymbol{x}) = \boldsymbol{\phi}_r(\boldsymbol{w}) \tag{47}$$

where

$$\phi_r(\boldsymbol{y}-\boldsymbol{x}) = \begin{bmatrix} 1 \\ y_1 - x_1 \\ y_2 - x_2 \\ (y_1 - x_1)(y_2 - x_2) \\ (y_1 - x_1)^2 \\ (y_2 - x_2)^2 \end{bmatrix} = \begin{bmatrix} 1 \\ y_1 - x_1 \\ y_2 - x_2 \\ y_1 y_2 + x_1 x_2 - y_1 x_2 - y_2 x_1 \\ y_1^2 + x_1^2 - 2y_1 x_1 \\ y_2^2 + x_2^2 - 2y_2 x_2 \end{bmatrix}, \phi_r(\boldsymbol{w}) = \begin{bmatrix} 1 \\ w_1 \\ w_2 \\ w_1 w_2 \\ w_1^2 \\ w_2^2 \end{bmatrix} \tag{48}$$

We then have the extended system

$$\boldsymbol{A}(\boldsymbol{y})\phi_r(\boldsymbol{x}) = \boldsymbol{v}, \tag{49}$$

where

$$\boldsymbol{v} = \begin{bmatrix} 1 \\ w_1 \\ w_2 \\ w_1 w_2 \\ w_1^2 \\ v_2^2 \end{bmatrix}, \phi_r(\boldsymbol{x}) = \begin{bmatrix} 1 \\ x_1 \\ x_2 \\ x_1 x_2 \\ x_1^2 \\ x_2^2 \end{bmatrix}, \boldsymbol{A}(\boldsymbol{y}) = \begin{bmatrix} 1 & 0 & 0 & 0 & 0 & 0 \\ 0 & 1 & 0 & 0 & 0 & 0 \\ 0 & 1 & 0 & 0 & 0 \\ 0 & y_2 & y_1 & -1 & 0 & 0 \\ 0 & 2y_1 & 0 & 0 & -1 & 0 \\ 0 & 0 & 2y_2 & 0 & 0 & -1 \end{bmatrix}. \tag{50}$$

## H More Numerical Results

### H.1 Update Steps

We compare the update steps in the linear system (**??**) using the noise model (14) with different orders of moments constraints $r$ in Figure 6. We find that with higher moments, the density is more concentrated and converges faster to the ground truth.

### H.2 Prediction Steps

We only apply the prediction steps to the system on the matrix Lie group:

$$\boldsymbol{Z}_{k+1} = \boldsymbol{Z}_k \boldsymbol{U}_k \boldsymbol{W}_k \in \mathrm{SE}(2), \tag{51}$$

and compare the recovered distributions with Monte Carlo simulations. We find that with larger $r$, we can better recover the distribution. As marginalizing the distribution $p(\boldsymbol{x}_k|\boldsymbol{\lambda}_k^-)$ requires symbolic integration that is generally intractable, we apply the Langevin dynamics on $\mathrm{SE}(2)$ to sample and visualize the position part of MED. The MED is sampled by Langevin dynamics and visualized in $x - y$ plane in Figure 7.

### H.3 Comparison with Particle Filter

We compare MEM-KF with particle filters with $N = 1e1, 1e2, 1e3, 1e4$ and $1e5$ particles. For both cases in Section 7.3, we sample the max entropy association strategy to avoid overconfidence in the observation models. We consider the threshold for resampling as $0.5$ among all the trials. The estimated trajectories of all the cases are shown in Figure 5. The box plot statistics for the position error is shown in Figure 8.

We find that as more particles are considered, the particle filters converge to better solutions. In terms of numerical results, MEM-KF has compatible solutions with a slightly larger median error than the particle filters with $N \geq 1e3$. However, we note that the MEM-KF is a deterministic method that maintains a smooth analytical belief representation that is convenient for the extraction of global point estimation. Such a property is not possible for particle filters that are based on random sampling.

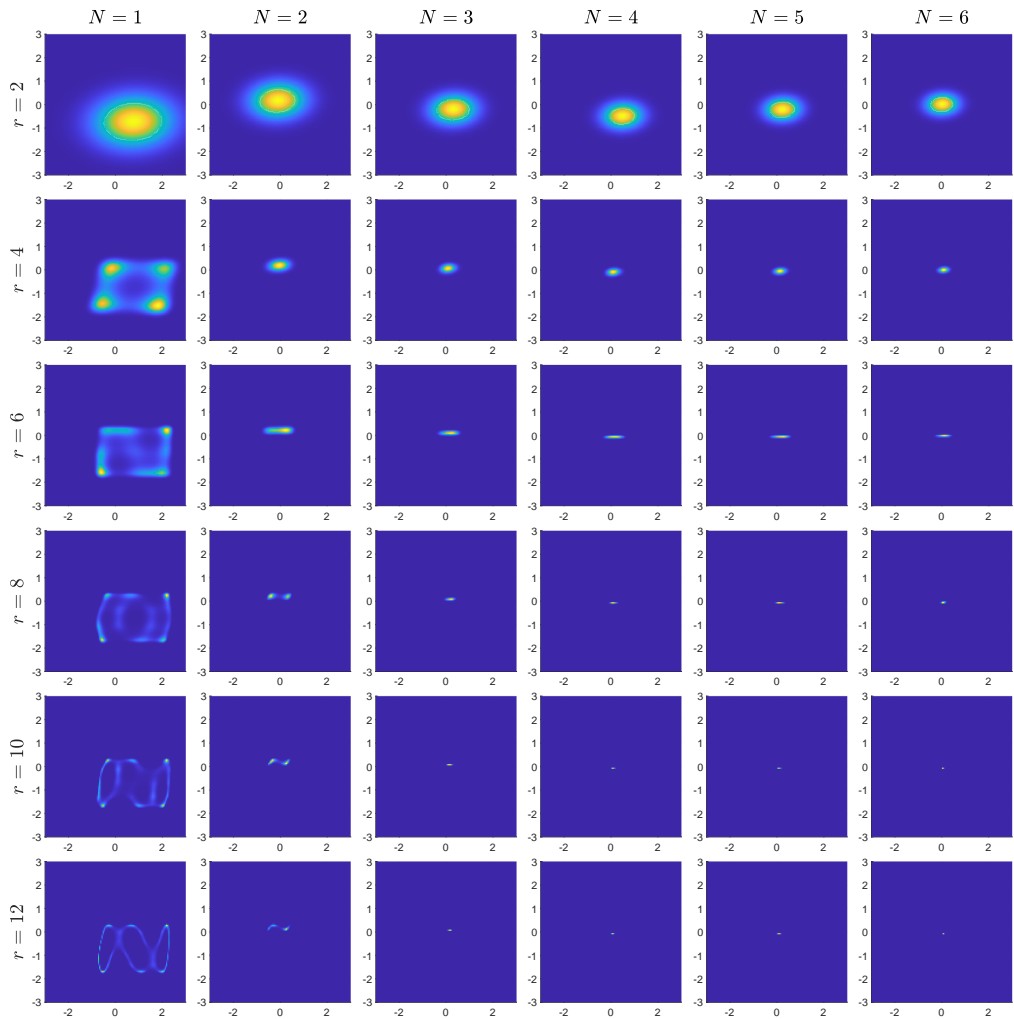

Figure 6: From the top to bottom, using noise model with moments with order up to $r = 2, 4, 6, 8, 10$ and 12 are matched. As more measurements are accumulated, the belief converges to the true state. With larger $r$, the belief converges much faster to the Dirac measure.

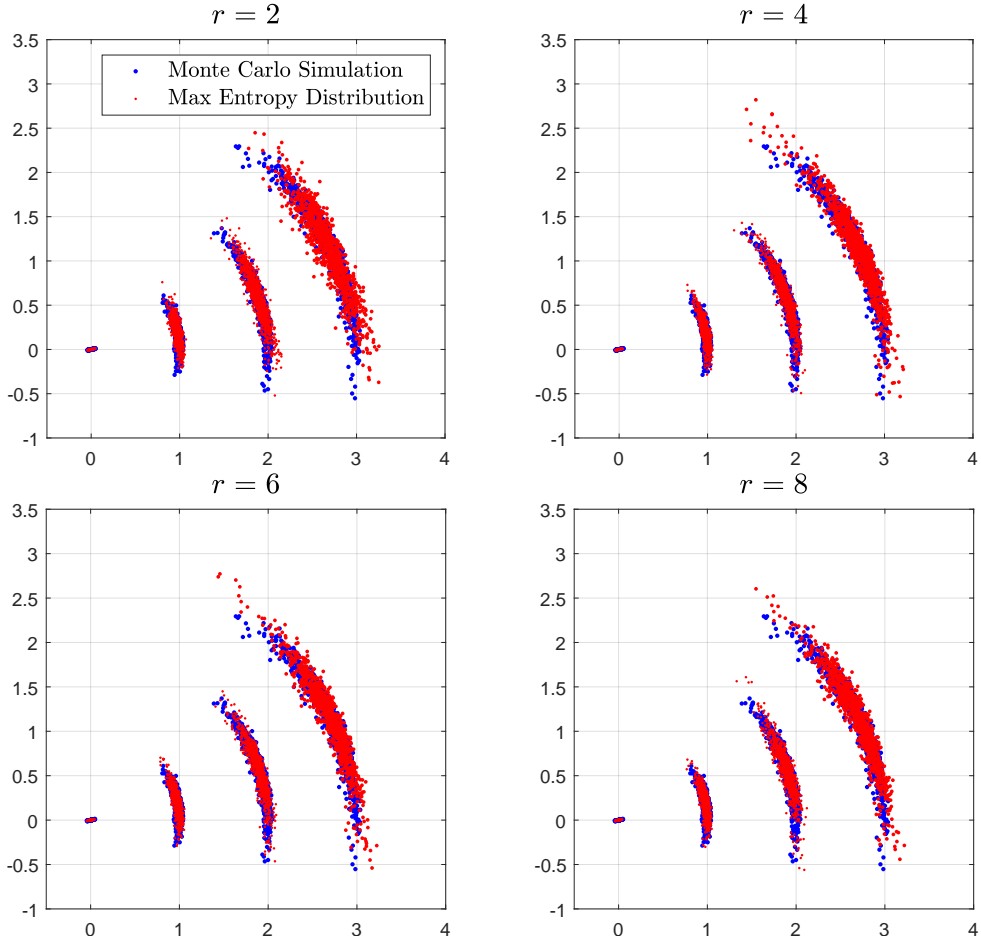

Figure 7: The propagation of the belief on SE(2) with only prediction steps. We visualize the position part in the $x - y$ plane. The blue dots represent the Monte Carlo simulations of the system dynamics, and the red dots represent the MED sampled by Langevin dynamics using the log likelihood of the recovered distribution. We find that with $r = 2$ the banana shape [47] is not fully captured by the MED. With larger $r$, the MED better captures the underlying distribution, especially in the tail of the banana shape.

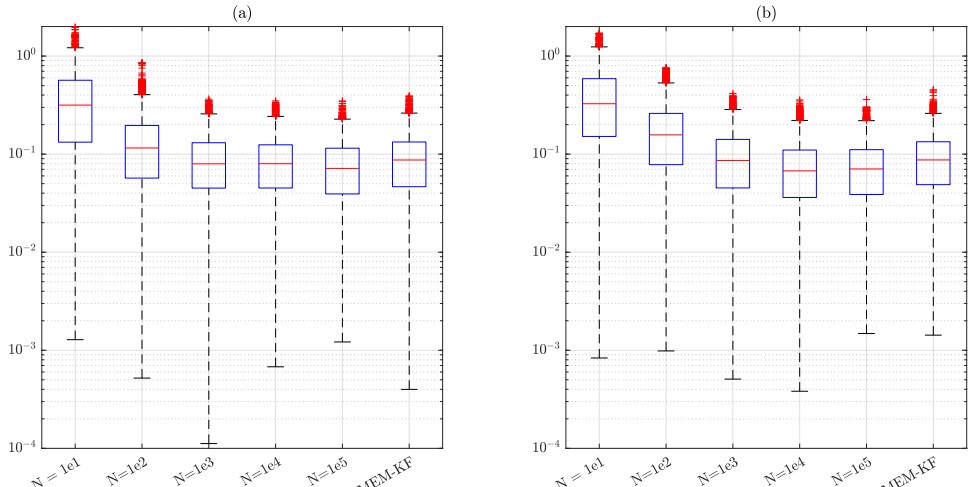

Figure 8: Location error for the localization tasks with unknown data association. (a)The underlying real association follows the same distribution in (17). (b)The measurement comes more from the landmarks at the lower left corner. We compare the 2-norm between the estimated position and the ground truth for all 100 trajectories. We compared the particle filters with $N$ particles with the proposed MEM-KF with polynomial order up to $r = 4$. We find that the particle filter with a sufficient number of particles and the proposed MEM-KF can estimate the trajectories with similar quality.

