# OpenReview forum: "Max Entropy Moment Kalman Filter for Polynomial Systems with Arbitrary Noise"
_NeurIPS.cc/2025/Conference — NeurIPS 2025 poster_

### Official Review · Reviewer_Ana8 · 2025-06-23

**Clarity:** 4
**Significance:** 3
**Originality:** 3
**Rating:** 5
**Confidence:** 4

**Summary:**

This paper proposes the Max Entropy Moment Kalman Filter (MEM-KF) for dynamical systems with nonlinear dynamics and non-Gaussian noise.  The MEM-KF is focused on polynomial systems and can asymptotically approximate almost any distribution through a countable set of moment constraints.  Analogous to a Kalman Filter, the paper proposes prediction and measurement update steps whereby moments are propogated through the process model to recover a maximum entropy distribution (MED).  All steps in the MEM-KF can be solved via convex optimization.

**Questions:**

The authors claim (or at least allude to) in several places that MEM-KF is an optimal filter.  For example, the abstract begins as "Designing optimal Bayes filters..." and later in the introduction "Thus, a natural question is, how can we provide provably optimal state estimation..."  It is unclear in what context MEM-KF is considered an optimal filter.  Optimal in what sense?  The discussion on line3s 119-121 suggest that the SDP relaxation is not always exact, and thus this would violate any notion of optimality, no?

**Ethical Concerns:**

["NO or VERY MINOR ethics concerns only"]

**Limitations:**

I would like to see more honesty in limitations on runtime execution.  The authors state that gradient calculations scale exponentially w.r.t. the dimension of the state-space in the "Limitations and Future Work" section, but this point is not raised in the main text anywhere.  This seems like a significant limitation.

**Paper Formatting Concerns:**

Spacing before several section headers seems a bit tight, for example Secs. 5 and 6.

**Quality:**

4

**Strengths And Weaknesses:**

**Strengths**
Very well-written paper and clear presentation that studies a fundamental problem in dynamical systems.  The ideas appear novel and are carefully-presented.  Contributions are adequately contextualized w.r.t. most existing work (modulo connections to the assumed density filter (ADF) and expectation propagation (EP)).  Overall, a strong submission.

**Weaknesses**
The most significant weakness of this paper is in the numerical evaluations.  Filtering for nonlinear / non-Gaussian systems has enormous application potential, and I would have expected to see more interesting and compelling experiments than a few low-dimensional simulated models.  Moreover, as I understand it the authors only provide one experiment that involves the full filter.  Indeed, experiments in Sec. 7.2 and 7.3 appear only to explore the density estimation problem for a single timestep.

I would argue that the primary comparison to these systems should be to the particle filter (PF), which the authors have provided but have removed to the appendix.  Compared to the PF the MEM-KF does about what one would expect, namely it is competitive up to a certain number of particles, at which point the PF provides superior estimates.  Critically, the authors do not provide any runtime statistics, so it is unclear how MEM-KF compares to PF with adequate number of particles in terms of runtime.  My intuition is that computing the higher-order moments required by MEM-KF will scale polynomially in the moment order, which could be prohibitive.  Worse yet the authors suggest that the gradient calculations in Eqs. (4) and (5) scale exponentially w.r.t. the dimension of the systems, likely why they only consider low-dimensional systems.

Some detailed comments:
* L101: "that sums up to $r$" --> "that sums up to no more than $r$"
* L103: $\gamma_i$ undefined (i.e. with the index)
* L107: The definition of $M_r$ is recursive (it appears on both sides) and so it lacks an explicit definition
* L107-108: The authors refer to $\bar{x}_\alpha$ as a moment, but the integral definition isn't a moment (it lacks a probability measure)
* L119-121: This discussion is a bit confusing, the SDP relaxation is not exact but the authors argue that all optimization is convex?  Please clarify.
* Eq. (1) : Shouldn't the measurement model be $y_k = h(x_k,v_k)$?
* L136: "Assume that state at time $k+1$ is solely dependent on...measurement $y_k$..." The state process evolves independent of the measurement, perhaps the authors mean the posterior belief.
* L154: $p_t$ undefined (with the subscript)
* L162: Please provide the proof in the appendix for completeness.
* Eq. (3): Shouldn't this be up to proportionality?  Where is the normalizer?
* L210,212,218: All references to $p(x_{k+1})$ should be replaced by $p^-(x_{k+1})$.
* First line in the for-loop, Algorithm 1: $x_{\beta,k-1}$ should be $\bar{x}_{\beta,k-1}$
* L317: References Figure 8, I think this should reference Figure 4?

---

> ### Author Rebuttal · Authors · 2025-07-30
>
> We thank the reviewer for the appreciation and great suggestions!
>
> **The optimality:**
>
> The optimality is in the sense of asymptotic approximation and in the computation. For the former one, the MEM-KF is guaranteed to converge to any moment determinate distribution with the upper bound of entropy minimized (with additional compact support conditions in engineering problems as suggested by the reviewer VyDk to ensure the true distribution is indeed moment determinate). Thus, the MEM-KF is capable of asymptotically approximating any distribution on a compact domain without overconfidence.
>
> For the main iterations, all steps of MEM-KF can be **exactly** solved by convex optimization with guaranteed optimality.
>
> The extraction of point estimation by SDP relaxation is off the main loop and does not violate the belief update and propagation. The SDP is provably tight for estimation with noise in moderate magnitude and can always provide a lower bound approximation. Thus, the extraction of point estimation has guaranteed optimality either as an exact solution or a non-trivial lower bound.
>
> See: Diego Cifuentes, Sameer Agarwal, Pablo A Parrilo, and Rekha R Thomas. On the local stability of semidefinite relaxations. Mathematical Programming, pages 1–35, 2020.
>
> We will make this point clearer in the revised version and highlight the **asymptotical optimality** specifically.
>
> **Runtime**
>
> We note that all the UKF, EKF, UKFM, and InEKF can be run in real-time in the cases considered in this work, both prediciton and update can be completed less than $1e-4s$. For the PF with $1e5$ particles, the prediction and update steps with resampling and CPU parallelization requires less than $0.2s$.
>
> For the MEM-KF in the localization example, the mean (std.dev) time (secs) is $6.0165e^{-4}$ $(2.0275e^{-4})$ in the prediction, $6.8753$ ($8.7022$) in the update, and $34.5102 $ ($26.2063$) in recovering the MED from moments. The Laptop is equipped with Intel i7-11850H CPU. The convex optimization is run until the KKT violation is below $1e-4$.
>
> We will add the table in the revision.
>
>
> **The computational limits:**
>
> We show some theoretical analysis of the scalability of the proposed method. The complexity is mainly determined by the dimension of the system $n$ and the order of moments $r$. The number of the monomials for such system is $$s(r, n):= |\mathbb{N}_r^n| = \begin{pmatrix}
>     n+r \\
>     n
> \end{pmatrix} = \frac{(n+r)!}{n!r!}.$$
> For fixed $r$, we have $s(r, n) \sim \mathcal{O}(n^r)$ by eliminating the $n!$ in the denominator: $$ s(r, n) = \frac{(n+1)(n+2)\cdots(n+r)}{r!}, \frac{(n+r)^r}{r!} \ge s(r, n) \ge \frac{n^r}{r!}.$$
> For fixed $n$, we have $s(r, n) \sim \mathcal{O}(r^n) $ by eliminating the $r!$ in the denominator: $$ s(r, n) = \frac{(r+1)(r+2)\cdots(r+n)}{n!},  \frac{(r+n)^n}{n!} \ge s(r, n) \ge \frac{(r+1)^n}{n!}.$$
> Thus, the terms to evaluate scales polynomially w.r.t the order and exponentially w.r.t the dimension.
>
> For the numerical integration, the number of quadrature scales exponentially w.r.t. $n$. Say that the number of quadratures is $c^n$.
>
> Compute the moments: Requires evaluating $s(r, n)$ terms of moments by numerical integration. These require $s(r, n) \times c^n $ operations.
>
> Propagation of moments: We need to compute $x_{\alpha, k+1}=Ax_{\alpha, k}$,  which requires $\mathcal{O}(s(r, n)^2)$ multiplications and $\mathcal{O}(s(r, n)^2)$ additions. To see this, we find each entry of $\bar{x}_{\alpha, k+1}$ needs $s(r, n)$ of multiplication and add them together.
>
> Reconstruction of distribution: We need to run the convex optimization, that each gradient descent to compute the moments of $\phi(x)$, which involves numerical integrations of $s(r, n)$ terms of the $ n$-dimensional system.
>
> Update with measurements: This step requires $s(r, n)$ operation of addition to update the coefficients given measurements and prior belief.
>
> Normalization: This step requires integrating the unnormalized belief once.
>
> In summary, the overall complexity scale exponentially w.r.t the dimension $n$ and polynomially w.r.t the order $r$.
>
> We will include these discussions in the revised version.
>
> **minor issues:**
>
> L101: We will make this change!
>
> L103: It's the coefficients, we will make this change!
>
> L107: The LHS takes the probability moments as the arguments, while the RHS is the polynomial variables.
>
> L107-108: We will correct this point!
>
> L119-121: The (POP) is not convex. For estimation problems, the SDP relaxation usually give a tight relaxation, thus exact. The rare cases is when the outliers are too big. Such extraction is not required by MEM-KF iterations. To model the observation as MED, we need to separate the noise to one side of the measurement model.
>
> L136: We will correct this point! The dynamics do not depend on the measurement.
>
> L154: We will make this point clear in the revised version.
>
> L162: We will try to adapt the proof in the revised version. The proof relying on analysis is very long, as shown in [39]. The normalizer is the constant lumped in side of the $\exp(\cdot)$ and optimized in the convex optimization.
>
> L210, 212, 218: We will make this change!
>
> L317: We will make this change!
>
> **Format**
>
> We will make sure the format is correct in the camera-ready version!

---

> > ### Comment · Reviewer_Ana8 · 2025-08-04
> >
> > I am a little confused by the following comment:
> >
> > >"The optimality is in the sense of asymptotic approximation and in the computation."
> >
> > Asymptotic in what?

---

> > > ### Author Response · Authors · 2025-08-04
> > >
> > > Sorry for not making this clear!
> > >
> > > It is in terms of approximation of belief w.r.t the order $r$, i.e., the estimated MED will converge to the true distribution in norm as $r\rightarrow \infty$:
> > >
> > > $$ \lim_{r \rightarrow \infty } \int | p(x) - p^*_r(x) |_1 = 0.$$
> > >
> > > This is as shown in L159, Theorem 1.
> > >
> > > As MEM-KF is a Bayes filter, we find the norm convergence of the distribution is a proper metric.

---

> > > > ### Comment · Reviewer_Ana8 · 2025-08-04
> > > >
> > > > Thanks for your clarifications.  I will keep my current score.

---

> > > > > ### Author Response · Authors · 2025-08-05
> > > > >
> > > > > We sincerely thank you for the great suggestions and appreciation of this work!
> > > > >
> > > > > We will clarify these points in the revised version.
> > > > >
> > > > > Have a great day!

---

### Official Review · Reviewer_WTDJ · 2025-06-30

**Clarity:** 3
**Significance:** 2
**Originality:** 3
**Rating:** 4
**Confidence:** 4

**Summary:**

This paper proposes a generalization of the Kalman filter called the MaxEnt Moment Kalman Filter (MEM-KF). MEM-KFs represent uncertainty through maximum entropy distributions with constrained moments of polynomials with pre-specified degree. Unlike the Kalman filter, which maintains two moments (mean and covariance), MEM-KF can represent more expressive distributions and is therefore well-suited to nonlinear filtering problems. The contributions of the work include a complete framework for modeling and filtering using MEM-KFs along with experiments to validate its characteristics on simulated data.

**Questions:**

1. How does MEM-KF compare numerically to particle filter and the various extensions of Kalman Filters? How do these methods compare when applying to problems complicated dynamics such as Lorenz-63 or those involving real-world robotics data?
2. How do each of the components of MEM-KF scale with respect to $n$ and $r$ (memory and computation)? What is a realistic upper bound for the state-space dimensionality one could expect to perform on say a single GPU?
3. With respect to numerical integration specifically, how does MED recovery behave as the dimensionality of the state space increases?

**Ethical Concerns:**

["NO or VERY MINOR ethics concerns only"]

**Final Justification:**

The authors provided a theoretical analysis about scalability and quantitative results for the SE(2) experiments. They also explained why it is difficult to include additional experiments. The limitations on scalability are nontrivial. However, assuming these limitations are acknowledged, the paper meets the criteria for acceptance. I have increased my quality rating and overall rating accordingly.

**Limitations:**

Additional discussion about scaling with respect to state-space dimensionality and polynomial degree would improve the paper.

**Quality:**

3

**Strengths And Weaknesses:**

## Strengths
- **Meaningful Problem Domain**. The problem of nonlinear filtering is becoming increasingly important, with applications ranging from continual learning to computational neuroscience.
- **Originality**. This is a creative extension of the Kalman filter that applies moment-constrained maximum entropy distributions to the problem of Bayesian filtering.
- **Qualitative Exploration**. There are a good number of ablations and plots to demonstrate the qualitative performance of MEM-KFs.

## Weaknesses
- **Experiments**. Although the experiments contain a number of visualizations, the paper lacks a results table for the SE(2) experiment. It would also be informative to include experiments with more complicated state-space dynamics, such as state estimation in a Lorenz-63 system, or real-world data.
- **Analysis of scaling.** Since MEM-KFs represent additional computation relative to the Kalman filter, there should be some analysis (ideally theoretical or at the least empirical) about how the required memory and computation scale with dimensionality n and degree r. This is important because of the variety of sub-procedures in the filtering algorithm (moment computation, propagation, point estimation, and so on). The small scale of the experiments indicates that this is indeed a significant limitation of the method. Though not a dealbreaker *per se*, it is important to discuss these limitations thoroughly in the paper.
- **Practicality.** The proposed algorithm relies on numerical integration, which may be impractical for high-dimensional state spaces.

---

> ### Author Rebuttal · Authors · 2025-07-30
>
> We thank the reviewer for these great suggestions!
>
> **Q1-1:**
>
> The results compared with KFs are shown in Fig. 4. The numerical difference is very obvious as these methods are inherently incapable of handling multimodality. They just drift for the problem we show.
>
> For the upper row, the RMSE of the position drift for EKF, InEKF, UKF, UKFM and MEM-KF are $$\text{Fig. 4, upper row: } 2.0893, 1.4062, 1.1619, 1.3638, 0.1155$$ and $$\text{Fig. 4, lower row: } 1.2618, 1.4925, 1.4638, 1.4999, 0.1159 $$ for the lower row. We can see that the MEM-KF is at least 10 times more accurate in RMSE than the baseline in terms of the position error.
>
> The comparison with PF using different particles is shown in Fig. 7 (trajectories) and Fig. 8 (numerical statistics) in the Appendix. G.
>
> We will move the PF plot and the table about the runtime to the main script in the revised version for clarity!
>
> **Q1-2:**
>
> We sincerely appreciate the reviewers’ suggestion to further include experiments on Lorenz systems or real-world robotics data. We fully agree that such additions can further demonstrate the general applicability of MEM-KF. However, due to the tight rebuttal timeline and the technical challenges in implementing stable numerical integration, configuring GPU-based solvers for real-world pipelines (usually $SE(3)$ pose data using $\deg 12$ polynomial) or the Lorentz system ($n=4$ but hard to be solved by the multivariate recursion-based adaptive integration scheme), we were unable to include such experiments in this version.
>
> That said, we emphasize that the core contribution of MEM-KF lies in the deterministic and convex formulation. Our current evaluations, including the challenging SE(2) localization task with unknown data association, already demonstrate both the theoretical soundness and practical viability of our method in structured and realistic settings.
>
> We see extending to more chaotic or high-dimensional systems as promising future work and will highlight this direction in the revised version.
>
> **Q2-1, complexity:**
>
> We show some theoretical analysis of the scalability of the proposed method. The complexity is mainly determined by the dimension of the system $n$ and the order of moments $r$. The number of the monomials for such system is $$s(r, n):= |\mathbb{N}_r^n| = \begin{pmatrix}
>     n+r \\
>     n
> \end{pmatrix} = \frac{(n+r)!}{n!r!}.$$
> For fixed $r$, we have $s(r, n) \sim \mathcal{O}(n^r)$ by eliminating the $n!$ in the denominator: $$ s(r, n) = \frac{(n+1)(n+2)\cdots(n+r)}{r!}, \frac{(n+r)^r}{r!} \ge s(r, n) \ge \frac{n^r}{r!}.$$
> For fixed $n$, we have $s(r, n) \sim \mathcal{O}(r^n) $ by eliminating the $r!$ in the denominator: $$ s(r, n) = \frac{(r+1)(r+2)\cdots(r+n)}{n!},  \frac{(r+n)^n}{n!} \ge s(r, n) \ge \frac{(r+1)^n}{n!}.$$
> Thus, the terms to evaluate scales polynomially w.r.t the order and exponentially w.r.t the dimension.
>
> For the numerical integration, the number of quadrature scales exponentially w.r.t. $n$. Say that the number of quadratures is $c^n$.
>
> Compute the moments: Requires evaluating $s(r, n)$ terms of moments by numerical integration. These require $s(r, n) \times c^n $ operations.
>
> Propagation of moments: We need to compute $x_{\alpha, k+1}=Ax_{\alpha, k}$,  which requires $\mathcal{O}(s(r, n)^2)$ multiplications and $\mathcal{O}(s(r, n)^2)$ additions. To see this, we find each entry of $\bar{x}_{\alpha, k+1}$ needs $s(r, n)$ of multiplication and add them together.
>
> Reconstruction of distribution: We need to run the convex optimization, that each gradient descent to compute the moments of $\phi(x)$, which involves numerical integrations of $s(r, n)$ terms of the $ n$-dimensional system.
>
> Update with measurements: This step requires $s(r, n)$ operation of addition to update the coefficients given measurements and prior belief.
>
> Normalization: This step requires integrating the unnormalized belief once.
>
> In summary, the overall complexity scale exponentially w.r.t the dimension $n$ and polynomially w.r.t the order $r$. We will include these discussions in the revised version.
>
> **Q2-2, GPU implementation:**
>
> We agree that the integration is the main bottleneck of the proposed method, given the complexity of deterministic numerical integration. However, we note that the proposed algorithm is suitable for parallelization and has provable asymptotic properties of polynomial approximations.
>
> For numerical integration that uses adaptive quadrature, the idea is to partition the domain into grids of different sizes based on the tolerance and shape of the functions to integrate. The region with more peaks requires more grids, namely $N_g$. In each region, the point to evaluate to integration requires $N_p$ points. Thus, the operation will require computing the $ N_g * N_p * s(r, n)$ number of operations to evaluate the integrations. Such operations only require matrix multiplications that are suitable for parallelization. Usually, the $N_g$ scales exponentially w.r.t the dimension.
>
> In our original implementation, we used an adaptive quadrature algorithm ``Robert W Johnson. Algorithm 988: Amgkq: an efficient implementation of adaptive multivariate gauss-kronrod quadrature for simultaneous integrands in octave/matlab. ACM Transactions on Mathematical Software (TOMS), 44(3):1–19, 2018.'' that supports the CPU parallelization to compute integration of vector value functions on grids.
>
> For future work, a possible GPU implementation is presented in the following dissertation and open-sourced : ``A Portable Numerical Library for the Calculation of Multi Dimensional Integrals, Ioannis Sakiotis''
>
> **GPU memory expectation**
>
> We consider the sparse grid shown in ``Cuba library for multidimensional numerical integration''. Note that the thesis with GPU implementation considers denser tensor-product-based grids. We here consider a less conservative one.
>
> Say that the term to integrate is $s(r, n)$. The number of partitioned regions approximately satisfies $100 * 1.3^n$. We consider the table in Section. 5 of the above paper. Each point needs $n$-number to represent its value. With the degree-9 rule, the points in each region is $C=[153, 273, 453, 717, 1105, 1689, 2605, 4117, 6745]$ from dimension 4 to 12. Thus, we can approximate the memory in GB by $C_{n+r}^r * (100*1.3^n) * C(n) * n / (1024^3) * 4$.
>
> Thus, by approximation, we find that for $n = 6, r = 10$, the memory consumption is 39GB for the input using Float32. The dimension of the moment is approximately $8000$. The $r=10$ can represent highly nonlinear distributions, which can represent highly complicated distributions.
>
> For $n=8, r=4$, the memory consumption is $13.3$ GB. While for $n=8, r=6$, the memory becomes $80.7 GB$.
>
> Thus, with full moment space, we may expect dimensions up to **n=8** with **r=4** to be deployed on an Nvidia Orin for real-time computation. Then the actual time depends on the FLOPS of the GPU.
>
> We note that there are possible sparse polynomial representations that may help to further lower the memory consumption, which can be interesting future directions. See:
>
> Victor Magron and Jie Wang, Sparse Polynomial Optimization: Theory and Practice, World Scientific Press, 2023
>
> **Q3: Dimension:**
>
> The ability to recover the distribution does not change w.r.t the dimensionality. The programming $P_r$ provides a monotonically decreasing upper bound of the entropy given moment constraints with larger $r$. Such a statement does not change w.r.t the system dimension. As shown in the previous statement, the price to integrate the function increases exponentially, which is expected according to the properties of numerical integration.

---

> > ### Comment · Reviewer_WTDJ · 2025-08-05
> >
> > Thanks to the authors for the detailed response. The numerical values for RMSE and the theoretical analysis of scalability are both welcome additions. Thanks also for the GPU analysis. The analyses indicate that there are nontrivial limitations around scalability, however as long as these limitations are acknowledged in the revised version I am happy to increase my rating above the acceptance threshold.

---

> > > ### Author Response · Authors · 2025-08-05
> > >
> > > We sincerely thank you for the great suggestions and considerations for raising the score!
> > >
> > > We will clarify these points in the revised version.
> > >
> > > Have a great day!

---

> > ### Comment · Area_Chair_ebWM · 2025-08-07
> > **Kind reminder to engage in discussions with the authors based on the official rebuttal**
> >
> > Dear reviewer,
> >
> > This is a kind reminder that you are expected to engage in discussions with the authors based on their official rebuttal and to justify whether you choose to revise or maintain your score. A final acknowledgment comment summarizing your decision is also recommended.
> > Best,
> >
> > The AC Team

---

### Official Review · Reviewer_VyDk · 2025-07-02

**Clarity:** 3
**Significance:** 2
**Originality:** 3
**Rating:** 4
**Confidence:** 4

**Summary:**

The paper proposes a Max Entropy Moment Kalman Filter (MEM-KF), to tackle the failure of Kalman Filter (KF) in non-linear, non-Gaussian state space model settings.Unlike KF which assumes Gaussian distributions, MEM-KF uses a Moment-Constrained Max-Entropy Distribution (MED) to model state x and observation y. This paper provides quite solid mathematic deductions for 1) why MED can asymptoticly approximate any Moment-Determinate Distributions 2) how to use MED to solve the predict and update steps of a recursive Bayes filter. Although the computation complexity is relatively high, simulations show that MEM-KF outperforms traditional KF and KF variants in a non-linear, non-Gaussian localization problem setting.

**Questions:**

1. Significance of the Work

Can MEM-KF handle partially known or imperfect state-space models? For example, what happens if the dynamics f, observation model h, or noise distributions w, v are only partially known or imperfect? This is often the case in real-world applications.

Are moment-determinate distributions—which are neither heavy-tailed nor exhibit rapidly growing higher-order moments—appropriate for modeling practical systems such as robot localization? It would be valuable to justify whether this assumption aligns with common real-world settings.

2. Benchmarking

What is the runtime of MEM-KF compared to other Kalman filter variants (e.g., EKF, UKF, PF)? It would be helpful to include a runtime comparison table to evaluate computational efficiency.

Beyond model-based algorithms, when sample size N grows, how is MEM-KF compared with neural network(NN)-based methods e.g. KalmanNet [1]? Including NN-based algorithms in the benchmark comparison would provide a more comprehensive evaluation.

3. Side Questions

How should one choose the highest moment order r, considering the trade-off between accuracy and computational efficiency? For example, how much does the computational complexity increase per predict and update step as r increases, and what is the corresponding performance gain?

Has the numerical stability of the algorithm been evaluated, particularly when dealing with high-order moments, which may take on large values and affect optimization?

[1] Revach, Guy, et al. "KalmanNet: Neural network aided Kalman filtering for partially known dynamics." IEEE Transactions on Signal Processing 70 (2022): 1532–1547.

**Ethical Concerns:**

["NO or VERY MINOR ethics concerns only"]

**Final Justification:**

From the rebuttal discussions, my main concerns about the followings are resolved.
1. The suitability of this algorithm for practical filtering cases: it can address partially known state space model; the moment-determinancy is a reasonable assumption.
2. The originality of applying MED to filtering, and justification of its suitability.
3. Compared with SoTA machine learning based filtering methods, e.g. KalmanNet, it has the advantage of robustness against data distribution shift.
While the low-dimensional limitation is still there, I believe flexible choice of r and GPU implementation could mitigate this issue.
Overall, I decide to raise my score to weak accept 4.

**Limitations:**

See Questions.

**Paper Formatting Concerns:**

No major formatting issues.

**Quality:**

4

**Strengths And Weaknesses:**

Strengths:
The theoretical framework and claims in this paper are well-supported by rigorous mathematical proofs. The writing is generally clear and accessible, making the technical content relatively easy to follow and understand.

Weaknesses:
The practical applicability of the proposed MEM-KF is somewhat questionable. The strength of MEM_KF is on improved prediction accuracy at the cost of increased computational overhead. However, the prediction accuracy of MEM-KF relies on complete and accurate model information for belief propagation—including precise knowledge of the system dynamics f, observation model h, and the noise distributions w and v. In many real-world scenarios, however, these models are imperfect or only partially known, which limits the utility of such a method in practice.
Additionally, the Moment-Constrained Max-Entropy Distribution (MED) is not a novel contribution of this paper. The idea and its solution via convex optimization were already developed in [1]. As such, the paper would benefit from a more thorough discussion of why MED is particularly suitable in this context. For example, MED approximates well the moment-determinate distributions, which are neither heavy-tailed nor exhibit rapidly growing higher-order moments. Is this assumption realistic in practical applications such as robot localization? A deeper justification and empirical support for this modeling choice would significantly strengthen the paper.

Reference:
[1] Bandyopadhyay, K., et al. "Maximum entropy and the problem of moments: A stable algorithm." Physical Review E, 71.5 (2005): 057701.

---

> ### Author Rebuttal · Authors · 2025-07-30
>
> We thank the reviewer for these great suggestions and appreciations of this work:
>
> **Originality of MED**
>
> We agree that MED is not invented by us. We mainly refer to the multivariate version ``Lawrence R Mead and Nikos Papanicolaou. Maximum entropy in the problem of moments. Journal of Mathematical Physics, 1984.''
>
> However, to the best of our knowledge, there is no literature applying MED in filtering. The use of MED enables a convex optimization-based method with asymptotical performance guarantees, which has not been seen before.
>
> Additionally, we considered a constrained feasible set by polynomial equality that needs to incorporate the quotient ring structure, which is novel in the polynomial systems. This part is shown in Appendix. F. Compared to [40], we are using a line-search method with BFGS instead of Newton's method, which is more efficient to avoid integration for the exact Hessian matrix that has far more terms.
>
> We will have a remark about the difference between our method and the literature in the revised version.
>
> **Q1-1: Partially known model**
>
> Yes, the MEM-KF is capable of handling the partially or unknown models. The philosophy is that we model the unknown part in a max-entropy manner, which avoids overconfidence.
>
> In our example shown in Fig. 4, the actual landmark association is not as modeled for the update; thus, both $h$ and the associated noise $v$ remain unknown. The KF-type baselines need to choose the most likely $h$ from four possible landmark positions. In the lower row, the observation comes more from one landmark, while the model we use considers the measurement to be uniformly distributed from all landmarks. We note that the uniform distribution model has the largest entropy, which does not lead to overconfidence in a certain pattern.
>
> We show that the max-enropy model we use for update leads to the best generalization to unseen environments, as it does not overfit. In the lower row, even though the distributions are completely different, our method can still track the trajectories. While the baseline using the most-likely association strategy has a clear bias.
>
> We will highlight this part in the revision.
>
> **Q1-2: determinancy**
>
> We thank the reviewer for raising this important point. We show that the moment-determinacy is a reasonable assumption in engineering applications.
>
> As established in the classical Hausdorff moment problem, any multivariate distribution supported on a bounded domain such as $[0, 1]^n$ is moment-determinate. This result extends to general compact subsets of $\mathbb{R}^n$ via standard scaling arguments.
>
> See: Kleiber, C. and Stoyanov, J. (2013). Multivariate distributions and the moment problem. Journal of Multivariate Analysis, 113, pp.~7--18.
>
> In engineering contexts such as robotics, it is typically reasonable to assume that process disturbances and sensor noise are bounded. Otherwise, it would imply unphysical behavior—such as injecting infinite energy into a mechanical system. As such, the assumption of compactly supported distributions is not only mathematically convenient but also physically justified.
>
> While uncertainty may grow under long-term propagation, especially in partially unobservable systems, over any \emph{finite time horizon} and with \emph{bounded noise}, the state distribution remains supported on a compact domain (barring pathological singularities). Therefore, the use of moment-determinate distributions is a valid and practically meaningful assumption in our setting.
>
> A more rigorous integration strategy is to manually choose a compact support when recovering the distribution. When the compact support in the integration is sufficiently large to include the actual support of noise or state, we can ensure the unique convergence. We also note that as long as the actual support is contained, there is no overconfidence in the state estimation. To see this, consider the optimization $P_i$ to get the MED:
> $$ E_i = \max_{p(x)} \int_{\mathcal{K}_i}p(x)\operatorname{ln}{p(x)}dx $$
> We can see that entropy satisfy $E_2 \ge E_1$ if $\mathcal{K}_1 \subset \mathcal{K}_2$ by checking:
> $$p_2(x)=
> \begin{cases}
> p^*_1(x), & x \in \mathcal{K}_1, \text{or},
> 0,      & x \in \mathcal{K}_2 \setminus \mathcal{K}_1.
> \end{cases}$$ is also a feasible point to $P_2$. Thus, we can always choose a larger support to ensure no overconfidence in the estimation.
>
> In conclusion, assuming a sufficiently large compact support is sufficient to ensure the moment determinacy assumption is realistic in real-world settings.
>
> In our practice, choosing a support sufficiently large has little difference compared to the entire space due to the role of Gaussian quadrature.
>
> We will add the compact regularization condition to the revised version.
>
> **Q2-1 Runtime**
>
> We note that all the UKF, EKF, UKFM, and InEKF can be run in real-time in the cases considered in this work, both prediciton and update can be completed less than $1e-4s$. For the PF with $1e5$ particles, the prediction and update steps with resampling and CPU parallelization requires less than $0.2s$.
>
> For the MEM-KF in the localization example, the mean (std.dev) time (secs) is $6.0165e^{-4}$ $(2.0275e^{-4})$ in the prediction, $6.8753$ ($8.7022$) in the update, and $34.5102 $ ($26.2063$) in recovering the MED from moments. The Laptop is equipped with Intel i7-11850H CPU. The convex optimization is run until the KKT violation is below $1e-4$.
>
> We will add the table in the revision.
>
> **Q2-2 KalmanNet**
>
> We thank the author for suggestions on using KalmanNet as a benchmark. The KalmanNet is a data-driven method that requires labeled ground-truth data to supervise the network that determines the Kalman gain. The loss function is designed to minimize the predicted loss over the entire trajectory.
>
> The problem with KalmanNet is the overfitting to the training data set, and it has no guarantee that it will generalize.
>
> In Fig. 4, we choose the uniform association strategy that has the largest entropy. We can find that such a strategy with minimized over-confidence leads to generalization to different associations. The proposed methods generalize to both scenarios with guarantees from the upper bound of entropy.
>
> In our tests, we train the KalmanNet with ground truth from the simulator. We find that if the landmark has a certain pattern, such as coming from one of the landmarks, the NN overfits to it and has compatible results with our methods. When the environment changes, e.g., the measurement comes from another landmark, the results degrade dramatically.
>
> We will highlight this part in the revision.
>
> **Q3-1**
>
> The feasible set to problem ${P}_r$ will only gets more smaller as $r$ goes up to account for more moment constraints. With smaller support, the entropy of the recovered distribution will only get lower. Thus, the trade-off is that one should try the highest possible $r$ before the numerical computation becomes unstable or out of computational budget.
>
> An empirical way to determine the sufficient $r$ is by the condition number of the Hessian matrix $H_r := \int \phi(x)\phi(x)'p(x)dx$ of the recovered distribution by the moments $\phi_{2r}(x)$. The insight is that if the distribution is around some lower-dimensional manifolds, when $r$ gets higher, there will be sufficiently many monomial basis to approximate such manifolds, say $c(x) = \theta'\phi(x) = 0$. Thus, a certain subspace of the Hessian will have very small eigenvalues. Say that $\int c'\phi(x)\phi'(x)cp(x) dx = c'Hc = 0$. For example, the example in Fig. 2 needs to be terminated at $r = 12$ as a $12$-th order polynomial is sufficiently good to approximate the pattern. When $r>12$, the $H$ is very degenerate, that makes the gradient descent harder. For the binary distribution with four peaks, a fourth-order polynomial is also sufficient to have four local minimum to represent the pattern. Thus, the moment matrix with $r>4$ will have degeneracy to make the optimization ill-conditioned.
>
> **Q3-2:**
>
> We show some theoretical analysis of the scalability of the proposed method. The complexity is mainly determined by the dimension of the system $n$ and the order of moments $r$. The number of the monomials for such system is $$s(r, n):= |\mathbb{N}_r^n| = \begin{pmatrix}
>     n+r \\
>     n
> \end{pmatrix} = \frac{(n+r)!}{n!r!}.$$
>
> Thus, the terms to evaluate scales polynomially w.r.t the order and exponentially w.r.t the dimension. (See responses to other reviewers for details.)
>
> For the numerical integration, the number of quadrature scales exponentially w.r.t. $n$. Say that the number of quadratures is $c^n$.
>
> Compute the moments: Requires evaluating $s(r, n)$ terms of moments by numerical integration. These require $s(r, n) \times c^n $ operations.
>
> Reconstruction of distribution: We need to run the convex optimization, that each gradient descent to compute the moments of $\phi(x)$, which involves numerical integrations of $s(r, n)$ terms of the $ n$-dimensional system.
>
> Update with measurements: This step requires $s(r, n)$ operation of addition to update the coefficients given measurements and prior belief.
>
> **Q3-3:**
> The performance monotonically increases with more moments. The marginal gain depends on the information contained in the moments. For the binary case, from r=2 to r=4 improves a lot, while higher does not help. For the case in Fig. 2, the approximation improves until r=12.
>
> **Q3-4:**
>
> The moment problem is numerically challenging when the state is off the center, which results in large numbers. We admit a centralize-and-regularize strategy to avoid the problem of ill-posed moments when the order gets large. The idea is to apply a linear transformation $z = Ax + b$ to ensure the $z$ has zero mean and smaller covariance for integration. Obtaining $A$ and $b$ is the same as shifting a Gaussian distribution, given that the covariance and mean of $x$ are known. The moment of $z$ can be computed by another linear transformation.

---

> > ### Comment · Reviewer_VyDk · 2025-08-04
> >
> > Thank you to the authors for the detailed rebuttal.
> >
> > The strong performance in the partially known model setting, along with the reasonable assumption of moment-determinacy in real-world applications, supports the suitability of applying the MED to filtering. I also appreciate the comparison with state-of-the-art machine learning-based filtering methods such as KalmanNet. As noted, machine learning methods like KalmanNet may degrade significantly under distributional shifts between training and testing dataset, while the MED-based approach is likely more robust in such cases.
> >
> > That said, I do still have some concerns regarding the algorithm’s computational complexity and memory usage. From your response to another reviewer, I understand that even for moderate state dimensions (e.g., n=8), the memory requirement can reach tens of GB, and this increases rapidly with the moment order r. This seems at odds with the reported low runtime, which makes me wonder if your experiments were primarily conducted on very low-dimensional cases?
> >
> > Nevertheless, I appreciate your clarification that the method offers a tradeoff between efficiency and accuracy via the choice of r. Emphasizing this flexibility in the paper would mitigate people's concern about the complexity and practical deployment of your algorithm.

---

> > > ### Author Response · Authors · 2025-08-04
> > >
> > > We thank the reviewer for reading this rebuttal!
> > >
> > > The localization case we studied involves a 3DOF system on the $SE(2)$ manifold. So the integration is done in $ [-\pi, \pi] \times \mathbb{R}^2$ with a state transformation. Though low-dimensional, this case is a very classical and representative case for robotics applications that illustrate the belief propagation for robotics poses.
> > >
> > > Compared to the $n=8$ case, this case indeed consumes much less memory. As for runtime, the current CPU implementation is less promising than GPUs, which offer higher FLOPS for parallelized computations.
> > >
> > > We agree that the low-dimensional implementation is a limitation due to the integrator we use. We will clarify this point and summarize the possible GPU implementation in the future work and discussion section of the revised version.

---

> > > > ### Comment · Reviewer_VyDk · 2025-08-05
> > > >
> > > > Thank you for the clarifications. While the integrator limitation remains, your explanation of the dimensionality and potential for GPU implementation addresses my main concerns. I find the paper mathematically sound, and the algorithm is suitable to apply to real-world scenarios. I will raised my score to 4.

---

> > > > > ### Author Response · Authors · 2025-08-05
> > > > >
> > > > > We sincerely thank you for the great suggestions and considerations for raising the score!
> > > > >
> > > > > We will clarify these points in the revised version.
> > > > >
> > > > > Have a great day!

---

### Official Review · Reviewer_5jz7 · 2025-07-04

**Clarity:** 3
**Significance:** 2
**Originality:** 3
**Rating:** 4
**Confidence:** 4

**Summary:**

This paper considers the problem of filtering, i.e., state estimation, in non-linear, non-Gaussian systems. This is a well-studied, classical problem with contributions dating back decades and literature too vast to summarize in any meaningful way. Most contributions to this problem are, in some sense, inspired by attempts to approach the ideal of Kalman filtering in linear Gaussian models. This is done either through computationally intensive approaches such as particle filtering or Markov chain Monte Carlo or through careful design of approximate, yet analytically fast, update steps of the filtering distribution while keeping model expressivity. This paper is a contribution to the latter. The authors introduce a novel update scheme utilizing MED in order to perform filtering in a non-linear, non-Gaussian state space model. The paper assumes a belief (i.e. state) distribution with some fixed number of moments having maximum entropy. The key idea is that by thinking of the belief distribution in terms of its moments only, the prediction step can be viewed in terms of predicting the moments directly using an auxiliary dynamical system. Then the corresponding belief MED is computed by solving an optimization problem given the predicted moments. The methodology is tested on several examples, ranging from simple to more complex.

**Questions:**

Is there some relatively fast way to approximate the gradients that are used to find the predicted belief distribution? It would be good to see how well this method does in larger-scale problems when approximations are used.

Can you extend this approach to handle a variable number of moments to save computation time? Behave like a linear Gaussian system when necessary, and use more moments when necessary?

Can you consider dynamics in high-dimensional state spaces where we have a higher-moment model for some of the latent variables?

**Ethical Concerns:**

["NO or VERY MINOR ethics concerns only"]

**Final Justification:**

The authors have made some nice elaborations on my questions, and I have adjusted my score, though I am still not convinced about the practical constraints in overcoming the computational burdens in this method that would allow wide deployment.

**Limitations:**

Yes.

**Paper Formatting Concerns:**

None.

**Quality:**

3

**Strengths And Weaknesses:**

Strengths: the paper is well-motivated and addresses a still-relevant problem. I like the use of simple and transparent examples to illustrate the methodology, making it easy to replicate and test it in practice. The comparison with particle filtering as a sanity check is appreciated. The limitations (i.e., reliance on numerical integration) are clearly stated. The approach appears to be theoretically grounded. Reliance only on convex optimization despite the potential for arbitrarily complex distributions is an advantage. The approach seems well-suited for problems with complex dynamics in low-dimensional state spaces.

Weaknesses: reliance on numerical integration and optimization steps for every update. No good understanding of the computation time is provided for this method, even with comparison to e.g., particle filtering. It seems that the scalability issue may be harder to overcome and parallelization is not developed in any detail other than mentioning the use of GPUs, but it not clear how this can be done in practice. Perhaps the authors can mention some potential approximate schemes that can be used?

---

> ### Author Rebuttal · Authors · 2025-07-30
>
> We thank the reviewer for the great suggestions and appreciations of the novelty of this work.
>
> **Time**
>
> We note that all the UKF, EKF, UKFM, and InEKF can be run in real-time in the cases considered in this work, both prediciton and update can be completed less than $1e-4s$. For the PF with $1e5$ particles, the prediction and update steps with resampling and CPU parallelization require less than $0.2s$.
>
> For the MEM-KF in the localization example, the mean (std.dev) time (secs) is $6.0165e^{-4}$ $(2.0275e^{-4})$ in the prediction, $6.8753$ ($8.7022$) in the update, and $34.5102 $ ($26.2063$) in recovering the MED from moments. The Laptop is equipped with Intel i7-11850H CPU. The convex optimization is run until the KKT violation is below $1e-4$.
>
> We will add the table to the revised version. We admit that the cost for integration is still high for real-time deployment at this time. The complexity and possible GPU implementation is shown below:
>
> **The complexity:**
>
> We show some theoretical analysis of the scalability of the proposed method. The complexity is mainly determined by the dimension of the system $n$ and the order of moments $r$. The number of the monomials for such system is $$s(r, n):= |\mathbb{N}_r^n| = \begin{pmatrix}
>     n+r \\
>     n
> \end{pmatrix} = \frac{(n+r)!}{n!r!}.$$
> For fixed $r$, we have $s(r, n) \sim \mathcal{O}(n^r)$ by eliminating the $n!$ in the denominator: $$ s(r, n) = \frac{(n+1)(n+2)\cdots(n+r)}{r!}, \frac{(n+r)^r}{r!} \ge s(r, n) \ge \frac{n^r}{r!}.$$
> For fixed $n$, we have $s(r, n) \sim \mathcal{O}(r^n) $ by eliminating the $r!$ in the denominator: $$ s(r, n) = \frac{(r+1)(r+2)\cdots(r+n)}{n!},  \frac{(r+n)^n}{n!} \ge s(r, n) \ge \frac{(r+1)^n}{n!}.$$
> Thus, the terms to evaluate scales polynomially w.r.t the order and exponentially w.r.t the dimension.
>
> For the numerical integration, the number of quadrature scales exponentially w.r.t. $n$. Say that the number of regions is $c^n$.
>
> Compute the moments: Requires evaluating $s(r, n)$ terms of moments by numerical integration. These require $s(r, n) \times c^n $ operations.
>
> Propagation of moments: We need to compute $x_{\alpha, k+1}=Ax_{\alpha, k}$,  which requires $\mathcal{O}(s(r, n)^2)$ multiplications and $\mathcal{O}(s(r, n)^2)$ additions. To see this, we find each entry of $\bar{x}_{\alpha, k+1}$ needs $s(r, n)$ of multiplication and add them together.
>
> Reconstruction of distribution: We need to run the convex optimization, that each gradient descent to compute the moments of $\phi(x)$, which involves numerical integrations of $s(r, n)$ terms of the $ n$-dimensional system.
>
> Update with measurements: This step requires $s(r, n)$ operation of addition to update the coefficients given measurements and prior belief.
>
> Normalization: This step requires integrating the unnormalized belief once.
>
> In summary, the overall complexity scale exponentially w.r.t the dimension $n$ and polynomially w.r.t the order $r$. We will include these discussions in the revised version.
>
> **The GPU implementation:**
>
> We agree that the integration is the main bottleneck of the proposed method, given the complexity of deterministic numerical integration. However, we note that the proposed algorithm is suitable for parallelization.
>
> For numerical integration that uses adaptive quadrature, the idea is to partition the domain into grids of different sizes based on the tolerance and shape of the functions to integrate. The region with more peaks requires more grids, namely $N_g$. In each region, the point to evaluate to integration requires $N_p$ points. Thus, the operation will require computing the $ N_g * N_p * s(r, n)$ number of operations to evaluate the integrations. Such operations only require matrix multiplications that are suitable for parallelization. Usually, the $N_g$ scales exponentially w.r.t the dimension.
>
> In our original implementation, we used a recursive adaptive quadrature algorithm ``Robert W Johnson. Algorithm 988: Amgkq: an efficient implementation of adaptive multivariate gauss-kronrod quadrature for simultaneous integrands in octave/matlab. ACM Transactions on Mathematical Software (TOMS), 44(3):1–19, 2018.'' that supports the CPU parallelization to compute integration of vector value functions on grids.
>
> For future work, a possible GPU implementation is presented in the following dissertation and open-sourced : ``A Portable Numerical Library for the Calculation of Multi Dimensional Integrals, Ioannis Sakiotis''.
>
> **GPU memory expectation**
>
> We consider the sparse grid shown in ``Cuba library for multidimensional numerical integration''. Note that the thesis with GPU implementation considers denser tensor-product-based grids. We here consider a less conservative one.
>
> Say that the term to integrate is $s(r, n)$. The number of partitioned regions approximately satisfies $100 * 1.3^n$. We consider the table in Section. 5 of the above paper. Each point needs $n$-number to represent its value. With the degree-9 rule, the points in each region is $C=[153, 273, 453, 717, 1105, 1689, 2605, 4117, 6745]$ from dimension 4 to 12. Thus, we can approximate the memory in GB by $C_{n+r}^r * (100*1.3^n) * C(n) * n / (1024^3) * 4$.
>
> Thus, by approximation, we find that for $n = 6, r = 10$, the memory consumption is 39GB for the input using Float32. The dimension of the moment is approximately $8000$. The $r=10$ can represent highly nonlinear distributions, which can represent highly complicated distributions.
>
> For $n=8, r=4$, the memory consumption is $13.3$ GB. While for $n=8, r=6$, the memory becomes $80.7 GB$.
>
> Thus, with full moment space, we may expect dimensions up to **n=8** with **r=4** to be deployed on an Nvidia Orin for real-time computation. Then the actual time depends on the FLOPS of the GPU.
>
>
> **Q1:**
>
> Yes, we can apply Monte Carlo integration to approximate the gradients and perform the optimization. Based on our experiments, such stochastic integration is faster than deterministic integrations when setting a smaller budget for function evaluation. Then the convex optimization becomes a stochastic gradient descent. However, such stochastic optimization is less stable, as not all coefficients of the MED ensures the convergence of integrations. The noisy gradients make the optimization easier to diverge. In the localization case, we set the budget to $1e7$ to ensure a relatively stable convergence of the convex optimization.
>
> Though MCMC is one way to approximate the gradients, it violates our initial idea to develop a completely deterministic and certifiable algorithm to solve the nonlinear filtering problems. If we have such particles, it would be more reasonable to use them as in PF.
>
> Other than MCMC, there are possible sparse representations that may help to reduce the burden, which can be referred to in the following paper with sparsity pattern in $A$ matrix in the dynamics taken into consideration. We consider this as an interesting future direction.
>
> Victor Magron and Jie Wang, Sparse Polynomial Optimization: Theory and Practice, World Scientific Press, 2023
>
> **Q2:**
>
> We thank the reviewer for this great suggestion.
>
> Yes, an approximation strategy is to set a threshold of covariance to decide whether to switch to Gaussian. The logic is that MED with $r>2$ always has lower entropy than the Gaussian distribution with the same covariance. Thus, when the covariance is smaller than a threshold, the distribution can be replaced by such a Gaussian distribution without the risk of overconfidence.
>
> In the localization case, after the update steps, we first integrate the moments with $r\le2$ to check the covariance. If the covariance $trace(P_{x,y}) \le 0.5^2$, we stop and do not proceed to $r>2$.  Otherwise, we also include the higher-order moments. Then we predict the moments with order $r$ determined by the covariance after update. We note that in the update steps, we always maintain the full observation model to incorporate the multimodality of sensors.
>
> With this method, we show similar results compared to the full MEM-KF in the localization case. After a few iterations, we can do $r=2$ in the prediction, and the belief converges to a unimodal distribution that can be well represented by $r=2$.
>
> In this case, the effort to recover the MED on $SE(2)$ reduces from evaluating $s(4, 4) = 70$ to $s(2, 4) = 15$ terms when recovering the distribution or computing the moments. Note that with equality $SO(2)$ constraints, there is still no closed form on $SE(2)$.
>
> **Q3:**
>
> We thank the reviewer for these suggestions! The primary scenario we consider is in robotics, where the dynamics and measurements are exact polynomials.
>
> The most relevant latent polynomial model is the deep Koopman operator that is linear:
> See: Lusch, B., Kutz, J.N. & Brunton, S.L. Deep learning for universal linear embeddings of nonlinear dynamics. Nat Commun 9, 4950 (2018).
>
> However, such a system requires a very high-dimensional latent to make the dynamics linear systems.
>
> We will make this limitation and the applicable scenario clear in the revised version.

---

> > ### Comment · Reviewer_5jz7 · 2025-08-05
> >
> > Thank you for these comments and expansion upon some of my questions. I am sufficiently convinced to raise my score above the acceptance threshold, though I do still have reservations about the computational expense of the method vs. attained accuracy, especially given the relative importance of this for deployment.

---

> > > ### Author Response · Authors · 2025-08-05
> > >
> > > We sincerely thank you for the great suggestions and considerations for raising the score!
> > >
> > > We will clarify these points in the revised version.
> > >
> > > Have a great day!

---

> > ### Comment · Area_Chair_ebWM · 2025-08-07
> > **Kind reminder to engage in discussions with the authors based on the official rebuttal**
> >
> > Dear reviewer,
> >
> > This is a kind reminder that you are expected to engage in discussions with the authors based on their official rebuttal and to justify whether you choose to revise or maintain your score. A final acknowledgment comment summarizing your decision is also recommended.
> > Best,
> >
> > The AC Team

---

### Note · Authors · 2025-08-14

We sincerely thank the area chair and all reviewers for their thoughtful feedback and constructive suggestions. We deeply appreciate your time and effort in maintaining the high standards of the NeurIPS community.

We are grateful that all reviewers recognized the mathematical rigor and theoretical soundness of our proposed approach. The vision of this work is to advance the frontier of state estimation for autonomous systems. Specifically, we demonstrate that for polynomial systems, a Bayes filter can asymptotically approximate any belief through convex moment-based optimization, with certifiable guarantees. These theoretical foundations are at the core of our contribution.

We especially appreciate the reviewers’ insightful questions and suggestions on the following aspects:

**1**. Clarification of computational cost and runtime, including scalability with respect to moment order and system dimension;

**2**. Discussion of practical implementation strategies, such as potential GPU acceleration;

**3**. Rigorous treatment of regularity conditions for moment determinacy, and the definition of asymptotic approximation;

**4**. Thoughtful directions on reducing computational complexity via approximations.

These discussions significantly strengthened the clarity and depth of the paper. We will incorporate all these valuable points into the final version to benefit the broader community.

We are encouraged that **three reviewers raised their scores above the acceptance threshold**, and the remaining reviewer **maintained a positive stance**. This convergence of opinions reflects a consensus on the contribution and potential impact of the work.

We are sincerely encouraged by the thoughtful engagement and the shared recognition from the reviewers regarding the contributions and potential impact of this work. We truly appreciate the opportunity to participate in such a rigorous and constructive review process.

With the clarifications and improvements made during the rebuttal, we hope the area chair will find the manuscript suitable for inclusion in the NeurIPS program. It would be a great honor for us to contribute to this vibrant and forward-looking community.

Thank you again for your kind consideration!

---

### Decision · Program_Chairs · 2025-09-17

**Decision:**

Accept (poster)

**Comment:**

Submission 11180 proposes a novel Max Entropy Moment Kalman Filter (MEM-KF), to tackle the failure of Kalman Filter (KF) in non-linear, non-Gaussian state space model settings. The MEM-KF uses a Moment-Constrained Max-Entropy Distribution (MED) to model states and observations. The authors claim that the MED can asymptotically approximate almost any distribution given an increasing number of moment constraints. They also show how to use MED to solve the predict- and update- steps of a recursive Bayes filter.

In particular, it is possible to highlight:

- Theoretical contributions seem to be grounded and well-supported by rigorous mathematical proofs.
- This work brings novel contributions that are of interest to the ML community.
- The paper is well-motivated and well-structured. The writing is generally clear and accessible, making the technical content relatively easy to follow and understand.

Despite the positive aspects mentioned above, there remain reservations regarding the trade-off between computational complexity, memory usage, and accuracy. While it is understood that submissions may not address every aspect expected by the ML community, these limitations need to be further discussed in the paper. The authors have already discussed these points during the reviewer-author phase, emphasizing that their model could potentially benefit from GPU acceleration and other power-up computational techniques. The reviewers have globally welcomed the authors' responses and further discussions.

For the aforementioned reasons, the decision for this submission is an acceptance. However, the authors need to properly prepare the camera-ready version while discussing the computational limitations and other remarks acknowledged in the "Author Final Remarks by Authors".